# Tsunami hazard in Lombok & Bali, Indonesia, due to the Flores back-arc thrust

Raquel P. Felix[1], Judith A. Hubbard[1,2], Kyle E. Bradley[1,2], Karen H. Lythgoe[2], Linlin Li[3,4] and Adam D. Switzer[1,2]

[1]Asian School of the Environment, Nanyang Technological University, Singapore

[2]Earth Observatory of Singapore, Nanyang Technological University, Singapore

[3]School of Earth Sciences and Engineering, Sun Yat-sen University, Zhuhai, China

[4]Southern Marine Science and Engineering Guangdong Laboratory (Zhuhai), Zhuhai, China

*Correspondence to:* Raquel P. Felix (raquelpi001@e.ntu.edu.sg)

**Abstract.** The tsunami hazard posed by the Flores back-arc thrust, which runs along the northern coast of the islands of Bali and Lombok, Indonesia, is poorly studied compared to the Sunda megathrust, situated ~250 km to the south of the islands. However, the 2018 Lombok earthquake sequence demonstrated the seismic potential of the western Flores Thrust when a fault ramp beneath the island of Lombok ruptured in two Mw 6.9 earthquakes. Although the uplift in these events mostly occurred below land, the sequence still generated local tsunamis along the northern coast of Lombok. Historical records show that the Flores fault system in the Lombok and Bali region has generated at least six ≥Ms 6.5 tsunamigenic earthquakes since 1800 CE. Hence, it is important to assess the possible tsunami hazard represented by this fault system. Here, we focus on the submarine fault segment located between the islands of Lombok and Bali (below the Lombok Strait). We assess modeled tsunami patterns generated by fault slip in six earthquake scenarios (slip of 1-5 m, representing Mw 7.2-7.9+) using deterministic modelling, with a focus on impacts on the capital cities of Mataram, Lombok and Denpasar, Bali, which lie on the coasts facing the strait. We use a geologically constrained earthquake model informed by the Lombok earthquake sequence, together with a high-resolution bathymetry dataset developed by combining direct measurements from GEBCO with sounding measurements from the official nautical charts for Indonesia. Our results show that fault rupture in this region could trigger a tsunami reaching Mataram in <9 minutes and Denpasar in ~23-27 minutes, with multiple waves. For an earthquake with 3-5 m of coseismic slip, Mataram and Denpasar experience maximum wave heights of ~1.6-2.7 m and ~0.6 to 1.4 m, respectively. Furthermore, our earthquake models indicate that both cities would experience coseismic subsidence of 20-40 cm, exacerbating their exposure to both the tsunami and other coastal hazards. Overall, Mataram city is more exposed than Denpasar to high tsunami waves arriving quickly from the fault source. To understand how a tsunami would affect Mataram, we model the associated inundation using the 5m slip model and show that Mataram is inundated ~55-140 m inland along the northern coast and ~230 m along the southern coast, with maximum flow depths of ~2-3 m. Our study highlights that the early tsunami arrival in Mataram, Lombok gives little time for residents to evacuate. Raising their awareness about the potential for locally generated tsunamis and the need for evacuation plans is important to help them respond immediately after experiencing strong ground shaking.

## 1 Introduction

Tsunamis sourced from back-arc thrust faulting, although not as common as megathrust tsunamis, could also result in fatalities and severe damage and destruction to structures. Such are the cases for the Mw 7.7 1991 Limon, Costa Rica (Suárez et al., 1995), Mw 7.9 1992 Flores Island, Indonesia, and Mw 7.5 1999 Ambrym Island of Vanuatu (Regnier et al., 2003) earthquakes. Understanding the tsunami hazard associated with back-arc thrusting is therefore important. Several studies have recognized the contribution of crustal earthquakes, which includes the back-arc thrusting, in the development of tsunami hazard assessments (Selva et al., 2016; Grezio et al., 2017; Behrens et al., 2021).

Here, we assess the deterministic tsunami hazard associated with the westernmost segment of the Flores Thrust, a back-arc thrust that extends for >1,500 km, accommodating a portion of the convergence between the Indo-Australian and Sunda Plates (Fig. 1a). Unlike its eastern segment, where the 1992 Mw 7.9 Flores Island earthquake occurred, the western part of the fault has not hosted devastating tsunamis in recent years, although historical records and previous studies show that it has generated at least eight tsunamigenic earthquakes (Fig.1b, NOAA database, Hamzah et al., 2000; Rastogi and Jaiswal, 2006; Musson, 2012; Nguyen et al., 2015, Tsimopoulou et al., 2020. The recent 2018 Lombok earthquake-triggered tsunamis were relatively minor because the earthquakes mostly occurred beneath the island itself and not offshore; nevertheless, the occurrence of the 2018 Lombok earthquakes gives new insights into the activity and geometry of this fault segment, and highlights the risk of earthquakes and associated tsunamis along strike.

Our study focuses on the tsunami hazard caused by slip on the Flores Thrust in the Lombok Strait, a 20-60 km-wide body of water between the islands of Lombok and Bali that connects the Java Sea to the Indian Ocean. Because of its geometry, slip on the thrust in the Lombok Strait could generate tsunamis that would efficiently propagate southwards and hit the west coast of Lombok and the east coast of Bali, where their capital cities (Mataram and Denpasar) are located.

### 1.1 Regional setting

Bali and Lombok islands, east of Java, are part of the Lesser Sunda Islands (Fig. 1a). They are located along the volcanic arc of the Java subduction zone, where the NNE-moving Indo-Australian Plate subducts beneath the Sunda Plate (Dewey and Bird, 1970; Hamilton, 1979; Bowin et al., 1980; Silver et al., 1983, 1986; Hall and Spakman, 2015; Koulali et al., 2016). The Java trench lies ~250 km to the south. The Flores back-arc thrust belt, on the other hand, follows the northern edge of the islands. Here, the kinematics of fault slip and folding are consistent with the sense of movement of the Indo-Australian Plate and associated shortening, indicating that the Flores back-arc thrust also formed to accommodate stress associated with the plate collision (Silver et al., 1983, 1986).

The Flores back-arc thrust is an east-west-trending, south-dipping fault zone that extends for >1,500 km along strike. It is composed of two main segments: the Wetar thrust zone to the east and the Flores Thrust to the west (Silver et al., 1983, 1986; Fig. 1a). From east to west, the Flores Thrust traverses just north of central Flores, Sumbawa, Lombok and Bali (Fig. 1a). From central Flores to east of Lombok, the thrust zone reaches to the

seafloor (Silver et al., 1983, 1986; Yang et al., 2020). As the deformation becomes blind from central Lombok to the west, the thrust zone has been mapped based on folds visible in seismic reflection data, and also manifests as a band of steeper north-facing slope on the seafloor (Silver et al., 1983; McCaffrey and Nabelek, 1987; Yang et al., 2020). West of Bali, folds are fewer and have no to little seafloor expression (Silver et al., 1983; Fig. 1d), suggesting that the Flores Thrust terminates at Bali (Yang et al., 2020). However, GPS measurements show that the north-south convergence rate in Bali (5 ± 0.4 mm/yr) is similar to that onshore Java (6 ± 1 mm/yr), therefore back-arc shortening may continue across a segment boundary along the Kendeng thrust in Java(Koulali et al., 2016).

**1.2 Seismicity of the Flores Thrust**

Focal mechanisms show that from February 1976 to February 2021, the Flores Thrust generated 29 Mw 5.5 to 7.8 earthquakes within the upper 40 km of the crust (GCMT; Fig. 1a). Earthquakes in this region can be caused by either tectonically driven fault slip or volcanic activity. In this back-arc region, most of the focal mechanisms are characterized by east-west striking nodal planes with a fault plane dipping 26±8°S; we infer that these are associated with the Flores Thrust.

The activity of this fault system is also testified by uplift recorded on its hanging wall. From eastern Sumbawa to central Flores, uplift is recorded by elevated terraces on the northern sides of the islands (Van Bemmelen, 1949). We suggest that the Quaternary reef terraces in northwest Bali (Boekschoten et al., 2000) are also related to tectonic uplift above the Flores thrust system, suggesting that the fault extends all the way to the western coast of the island (Fig. 2).

Although the earthquakes in this region are largely consistent with tectonic shortening, the active volcanoes not only generate their own seismicity, but also play a role in the horizontal and vertical distribution of fault-generated earthquakes (Lythgoe et al., 2021). A relationship between faulting and volcanic activity was observed for the 2018 Lombok earthquake sequence, which generated four >Mw 6 events between 28[th] July to 19[th] August. These earthquakes did not occur offshore on the northern frontal thrust of the Flores Thrust, but instead involved slip along the deeper part of the fault and associated imbricate thrusts beneath Lombok, to the north of the active Rinjani volcano (Salman et al., 2020; Yang et al., 2020; Lythgoe et al., 2021). While these earthquakes were not directly caused by volcanic activity, the presence of the volcano constrained the earthquake distribution by elevating the downdip limit of the seismogenic zone in the crust (Lythgoe et al., 2021). Based on relocated earthquakes and seismic reflection data analysis, the earthquakes occurred on the Flores fault ramp, a blind thrust dipping 25°S that flattens updip onto the Flores Thrust décollement at ~6 km depth (Lythgoe et al., 2021; Fig. 1c).

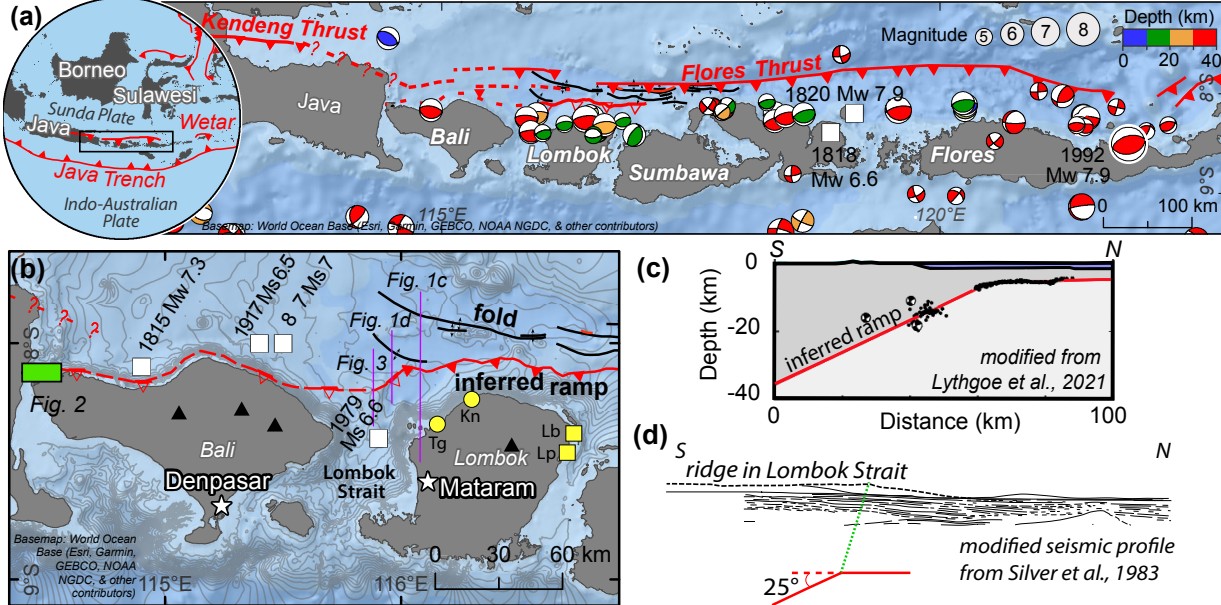


**Figure 1: Regional setting of the Flores Thrust and its subsurface ramp-flat geometry. (a) Circle - The Flores back-arc thrust system, which is located along the northern edge of the Lesser Sunda Islands. The thrust is composed of two segments: the Wetar thrust to the east and the Flores Thrust to the west (black rectangle). Seismicity (USGS earthquake catalogue, 1976-2021) and focal mechanism solutions (GCMT, 1976-2021) show that the Flores Thrust is seismically active. The Mw 7.9 Flores Island tsunamigenic earthquake is the largest earthquake on record for this system and occurred at the eastern end of the thrust. (b) The western part of the Flores Thrust has generated historical tsunamigenic earthquakes (white rectangles; www.ngdc.noaa.gov; Hamzah et al., 2000; Rastogi and Jaiswal, 2006; Musson, 2012; Nguyen et al., 2015; Griffin et al., 2019). Yellow squares and circles: towns where a tsunami was reported following the Mw6.4 28th July and Mw6.9 5th Aug 2018 events, respectively. Tg – Tanjung, Kn – Kayangan, Lb – Labuhan Pandan and Lp – Leper. We interpret that the blind ramp mapped at Lombok (Lythgoe et al., 2021) extends westwards based on the seafloor morphology and uplifted terraces in the northwestern part of Bali (green rectangle; Fig. 2). Basemaps – World Ocean Base. The map extent of (b) reflects the coverage of grid layer 1 (L1) used in the tsunami modelling. The basemap of (b) with only contour lines overlain is shown on Fig. S1. (c) The geometry of the blind fault ramp is constrained by the seismicity of the 2018 Lombok earthquake sequence (Lythgoe et al., 2021). (d) Gentle folds interpreted by Silver et al., (1983) based on a seismic profile across the Lombok Strait. Below the profile we show our inferred location for the fault ramp.**


**1.3 Tsunamigenic earthquakes of the Flores Thrust**
Historical records (NOAA database, www.ngdc.noaa.gov) and tsunami studies (Hamzah et al., 2000; Rastogi and
Jaiswal, 2006; Musson, 2012; Nguyen et al., 2015; Griffin et al., 2019) document at least four tsunamigenic
earthquakes on the Flores Thrust, in addition to the two earthquakes in 2018, which produced local inundation
(Fig. 1b) . Three of these events occurred in the western part of the thrust zone, north of Bali. The oldest event on
record is the 1815 Ms 7 earthquake, which triggered a landslide and tsunami; together, these events killed >1,200
people. NOAA categorizes this as a probable tsunamigenic event, as it is unclear whether the tsunami was caused
only by the coastal landslide, or by the earthquake and landslide together. The 1857 Ms 7 and 1917 Ms 6.5 events
are described by NOAA as definite and probable tsunamigenic earthquakes, respectively. The 1857 event
generated four consecutive tsunami waves, at least 3 m high, northwest of Flores Island (National Geophysical
Data Center / World Data Service: NCEI/WDS Global Historical Tsunami Database. NOAA National Centers for
Environmental Information. doi:10.7289/V5PN93H7). In addition, in the Lombok Strait, a 1979 Ms 6.6
tsunamigenic earthquake left 200 injured and killed 27 people, although the tsunami is poorly documented and
may have played a minor role in the destruction (Hamzah et al., 2000).

The best-documented tsunamigenic earthquake on the Flores Thrust occurred in its far eastern part
(Yeh et al., 1993; Imamura and Kikuchi, 1994; Tsuji et al., 1995; Pranantyo et al., 2021). The 1992 Mw 7.9 Flores
Island earthquake injured 2,144 people and killed 2,080 (Yeh et al., 1993; Tsuji et al., 1995; Fig. 1a). This
earthquake occurred at ~16 km depth (Beckers and Lay, 1995), and generated a tsunami that propagated to the
northern coast of Flores Island within five minutes (Yeh et al., 1993). Field mapping shows that the tsunami
inundated the land as far as 600 m, with an average run-up height of ~2 to 5 m (elevation reached above sea level).
Anomalously high run-up heights of 20-26 m to the northeast may be associated with submarine landslides (Yeh
et al., 1993).

The recent 2018 Lombok earthquake sequence occurred primarily below land, but nevertheless small-scale
tsunamis were reported by the residents of northern Lombok (Tsimopoulou et al., 2020). When the Mw 6.4 July
event occurred, the northern coast of Lombok subsided by ≤0.1 m (Wibowo et al., 2021), and the northeastern
coast was hit by a tsunami at the towns of Labuhan Pandan and Tanjung, which were inundated 10-70 m with
run-up heights of ~1-2.5 m. For the Mw 6.9 August 5 event, although the northern coast was uplifted by ≤0.5 m
(Wibowo et al., 2021), the residents of the northwest towns, Tanjung and Kayangan, reported a tsunami that
inundated 7-40 m inland with a run-up height of ~1.7-2 m (Fig. 1b).

Together, these records show that the Flores Thrust is capable of generating significant thrust earthquakes with
associated land uplift/subsidence as well as local tsunamis. The full tsunamigenic potential of this fault system is
not known, as the observational window is short compared to typical earthquake recurrence intervals. Here, the
observational window refers to the historical and seismic records. To our knowledge, there are no paleo-tsunami
studies in this area that are associated with the Flores Thrust. There is a paleo-deposit study in Bali, but it is
interpreted to be deposited by a tsunami generated by the megathrust rupture (Sulaeman, 2018). Hence, we rely
only on historical and seismic records when we refer to a short observational window. The tsunami studies related
to Flores Thrust are limited and they are about the numerical modelling of the historical tsunamis. Here, we
explore what could happen when coseismic slip occurs on the Flores thrust ramp within the Lombok Strait, and
how the generated tsunami and coseismic land deformation would together affect the coastal cities of Mataram,
Lombok and Denpasar, Bali.


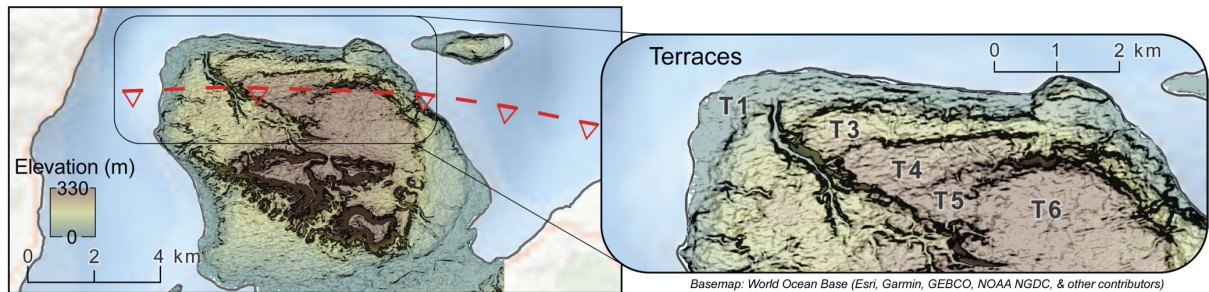

**Figure 2: Six coastal terraces (T1-T6) identified using a digital elevation model (DEMNAS) in northwest Bali, likely uplifted due to slip on the Flores Thrust ramp. The location is shown as a green rectangle on the map in Figure 1b. Basemap – World Ocean Base.**

### 1.4 Previous tsunami modelling studies

Tsunami modelling studies in this region commonly focus on the segment of the Sunda Megathrust along the Java trench (Okal and Borrero, 2011; Kurniawan and Laili, 2019; Suardana et al., 2019; Kardoso and Dewi, 2021) (Fig. 1a), with a few studies evaluating the western segment of the Flores Thrust (Rusli et al., 2012; Løvholt et al., 2012; Afif and Cipta, 2015), and four considering an earthquake sourced within the Lombok Strait (Rakowsky et al., 2013; Horspool et al., 2014; Pradjoko et al., 2018; Wibowo et al., 2021; Fig. 1b). All four studies show tsunami results in Mataram, Lombok; however, each study focuses on different aspects of tsunami modelling, and three predate the 2018 Lombok earthquake sequence, which illuminated important aspects of the fault geometry. The only study after the 2018 earthquakes (Wibowo et al., 2021) did not update their fault model to reflect new information about the geometry of the Flores Thrust derived from studies of the 2018 Lombok earthquake sequence. Overall, these prior results do not address the potential earthquake scenarios that we consider plausible: Rakowsky et al. (2013) study the sensitivity of inundation to land friction, Horspool et al. (2014) describe the probabilistic tsunami hazard, Pradjoko et al.(2018) considers a fault that is much too steep and uses bathymetry that is too coarse to produce reliable results, and Wibowo et al. (2021) did not consider the post-2018 earthquake studies of the fault geometry of the Flores Thrust.

Rakowsky et al. (2013) studied the sensitivity of inundation models in the region to the topography and friction parameters of the land surface. Their tsunami modeling was done using the ~900-m-resolution GEBCO dataset interpolated with measurements from ships and nautical charts; the interpolation method is not described in detail. They considered a Mw 8.5 earthquake and produced a maximum flow depth (vertical distance between the land and inundating water surface) of 10 m, with an inundation extent ranging from ~1-1.6 km in Mataram. This earthquake magnitude is larger than any observed event as the most recent estimates of the historical tsunamigenic earthquakes in the Flores Thrust ranges from Mw 6.6 to Mw 8.3, (Griffin et al., 2019), and that seismic records show that the 1992 Flores Island earthquake is Mw 7.9. They found that inundation distance depended on the topographic parameters: lower bottom friction or a bare earth digital terrain model produced higher inundation compared to higher friction or a digital surface model (with structures, e.g., houses). Their results highlight the importance of using an accurate surface model when assessing potential inundation.

Horspool et al. (2014) focused on probabilistic tsunami hazard for all of Indonesia. They used a bathymetry dataset
that combined GEBCO data with measurements from Navy charts and multibeam surveys. The maximum
magnitude calculated for the Flores thrust is Mw 8.1, Mw 8.3 and Mw 8.5 for fault dips of 25-27°. Their results
do not describe the regional hazard (e.g. wave heights, timing, inundation), but rather assess how much of the
local hazard is contributed by this fault system rather than the megathrust. They showed that for a 500-year return
period, the tsunami hazard in Mataram is 10-30% most likely due to the shallow part of the Flores Thrust.

Pradjoko et al. (2018) used a model of a Mw 6.4 earthquake to simulate a scenario similar to the 1979 event,
which was the largest recorded earthquake in this region prior to the 2018 Lombok earthquake sequence. They set
2.5 m of fault slip on a 72°-dipping fault (significantly steeper than the 25° dip we interpret for the fault) centered
at 25 km depth. Using GEBCO bathymetry to model tsunami propagation (with a coarse horizontal resolution of
~900 m), their results indicate that a Mw6.4 earthquake could generate a 0.13-0.2 m-high tsunami wave that
arrives at the coast of Mataram ~18-20 minutes after the earthquake.

The study by Wibowo et al. (2021) focused on the tsunami hazard posed by a Mw 7.4 earthquake on the Flores
thrust to the northern coasts of Lombok and Bali. They set 2.7 of slip on a 27°-dipping fault plane with dimensions
of 75 km x 27 km centered at 27 km depth. The fault parameters they used are based on the mean values of the
earthquake sources in the USGS 1900-2020 earthquake database. The orientation and depth of the fault are similar
to those we use in our modeling, but the updip tip of the fault in their model is located about 25 km north of the
islands rather than along the northern coast of the islands, as we interpret from the 2018 Lombok earthquake
sequence and bathymetry in the Strait. They used the 180-m resolution National Bathymetry of Indonesia
(BATNAS) dataset as input bathymetry in the numerical simulations. Their focus was on the impact along the
northern coasts, but they note that the tsunami arrives in Mataram and Denpasar in 9 and 25 minutes, respectively.
They also find that the maximum wave height is 1.5 m in Mataram and 1 m in Denpasar.

Following the 2018 Lombok earthquake sequence, we now have a more accurate understanding of the location
and subsurface geometry of the Flores Thrust in this region. Hence, the earthquake models we use in our study
are geologically well-constrained. In addition, since tsunami propagation in shallow water depends strongly on
the bathymetry, we develop and incorporate a new bathymetric model by combining the GEBCO dataset with
sounding measurements from the official nautical chart for Indonesia. This is particularly important along the
shallow coast, where seafloor roughness has a strong control on wave propagation. In our study, we show the
tsunami results from six different earthquake scenarios within the Lombok Strait, highlighting impacts on the
populated capital cities of Mataram, Lombok and Denpasar, Bali, as both cities face the Strait. We also calculate
the coseismic uplift and subsidence for varying slip amounts, and report this together with the tsunami time history
and pattern and the maximum wave height. An inundation scenario is also included for the city of Mataram.

**2 Methodology**
**2.1 Fault model setup**
The 2018 Lombok earthquake sequence illuminated the geometry of the Flores Thrust beneath Lombok (Fig. 1c).
Together, relocated aftershocks, earthquake slip distributions, and seismic reflection imaging indicate a blind fault
ramp dipping 25°S that flattens updip to a décollement at ~6 km depth and continues north below the Bali Sea.
The part of the thrust ramp that ruptured in the 2018 sequence extends 45 km downdip and 116 km lengthwise
(Lythgoe et al., 2021; Fig. 1c & 3).
We use these fault parameters to set up our fault model, choosing a fault with an east-west strike, similar to the
general trend of the Flores Thrust, positioned across the Lombok Strait. The complete parameters are listed in
Table 1. We are not trying to replicate the 2018 earthquakes, but rather consider an earthquake on the neighboring
part of the fault that did not rupture in that sequence. The eastern boundary of the fault model slightly overlaps
with the western limit of the 2018 earthquake sequence. Such overlapping ruptures have been observed in Kuril
Trench (Ammon et al., 2008) and Peru-Chile Trench (Bilek, 2010). We extend the western edge of the model to
below the eastern edge of Bali, in order to span the width of the Strait; the fault likely continues further west (as
evidenced by uplifted terraces and seismicity), but rupture to the west would occur below land and would not
contribute to a tsunami. As there are limited available information on the structural geology and the seismicity of
the Flores Thrust in this region, While there is limited data within the strait to assess the continuity of the fault,
there is no reason to believe that there are significant structural variations along strike. The focal mechanisms for
the events near Bali have very similar strike and dip to that at Lombok (Fig. 1a). When varying the fault dips to
18° and 34°, representing the minimum and the maximum limits of the fault dip uncertainty, they have minimal
impact on the tsunami model. The tsunami energies inherent in these two models are only 5-8% different from
the energy of our model with a 25° fault dip (Felix et al., 2021). Hence, minor structural variations would result
in minor changes in arrival times and wave heights but would not be likely to have a strong effect on our results.
**Table 1: Parameters of fault models A and B used in the numerical modelling.**

| Parameters | Fault model A | Fault model B |
|---|---|---|
| Epicenter longtitude | 115.77° E | 115.77° E |
| Epicenter latitude | 8.3821° S | 8.2905° S |
| Focal depth | 15.5 km | 10.8 km |
| Width | 45 km | 22.5 km |
| Length | 116 km ||
| Strike | 90° E ||
| Dip | 25° S ||
| Rake | 90° ||


We trace the upper blind tip of the fault ramp following the southern edge of a north-facing seafloor slope. This
surface morphology coincides with folding interpreted from seismic reflection surveys (Silver et al., 1983; Yang
et al., 2020), and we interpret that the folding formed due to slip across a bend at the upper tip of the blind fault
ramp (Fig. 1b). We extend the fault ramp to a depth of 25 km below the seafloor, which represents the maximum
seismogenic depth in this region based on historical seismic records and the maximum depth of seismicity
observed in the 2018 sequence (Lythgoe et al., 2021).
We model two fault ruptures on this fault (Models A and B, Fig. 3). Model A consists of a whole-fault rupture,
while Model B allows only the upper half of the ramp to slip. This second model represents a scenario similar to
the 2018 Lombok earthquakes, where most of the slip occurred on the shallow part of the fault ramp. However,
the maximum rupture depth at Lombok was interpreted to be limited by the elevated geothermal gradient
associated with the volcano. In the Lombok Strait, there is no such volcano; thus, it is likely that slip within the
Lombok Strait could reach deeper due to the colder geothermal gradient.

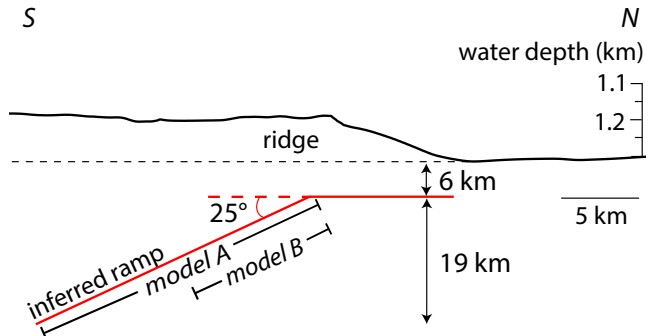


**Figure 3: Profile of the fault geometry used in the tsunami modelling relative to the seafloor ridge. We study two fault slip models: model A (whole-ramp rupture) and model B (slip only on the upper half of the ramp). The location of the profile is shown in Fig. 1b.**

**2.2 Slip model**
For both Models A and B, we consider three deterministic scenarios with uniform slip of 1, 3, and 5 m (six
scenarios total).The modeled historical tsunamigenic earthquakes in the Flores Thrust are estimated to have
magnitudes ranging from Mw 6.7 to Mw 8.5 (NOAA, Musson et al., 2019; Griffin et al., 2019). Using the
scaling relationship for magnitude and slip of shallow crustal reverse faulting by Thingbaijam et al. (2017),
these earthquake magnitudes have average slip ranging from 1 to 5 m. In order to represent this range, we use
the minimum (1 m), the mid-range (3 m) and the maximum (5 m) slip values in our modelling. In the
subsequent texts, we refer to these slip models as A-1, A-3 and A-5 for fault model A and B-1, B-3 and B-5 for
fault model B. We note that although modelling with more complex rupture scenarios would perhaps be a more
detailed option (e.g. Serra et al., 2021), the current information that we have about the Flores Thrust in Bali and
Lombok region, however, is limited. Hence, we think that it is better to use a planar fault model and uniform
slip to lessen the use of random parameters that could increase the uncertainty in the results. We also note that
although probabilistic approaches are becoming more common, the deterministic method is still included in
recent tsunami hazard studies (e.g. Wronna et al., 2015; Roshan et al., 2016; Gonzales, et al., 2019; Escobar et
al., 2020;  Rashidi et al., 2020;  Hussain et al. 2021; Rashidi et al., 2022).
In order to focus on the impact of tsunami generation, we include only slip on the fault ramp (no slip transferred
onto the northern décollement). This updip termination of slip was observed in the Lombok sequence (Lythgoe et
al., 2021) and is therefore realistic in our region to the west as well. Although we consider uniform slip, earthquake
slip is known to be spatially variable, and in particular to taper around the edges of the slip patch. We evaluate the
impact of this taper on the initial seafloor deformation using the Green's function for rectangular dislocations
(Okada, 1992) in the code Unicycle (Moore et al., 2019); we find that tapering the slip slightly modifies the uplift
profile by broadening it and shifting it to the south (downdip direction) but does not significantly change the
model (Fig. 4).

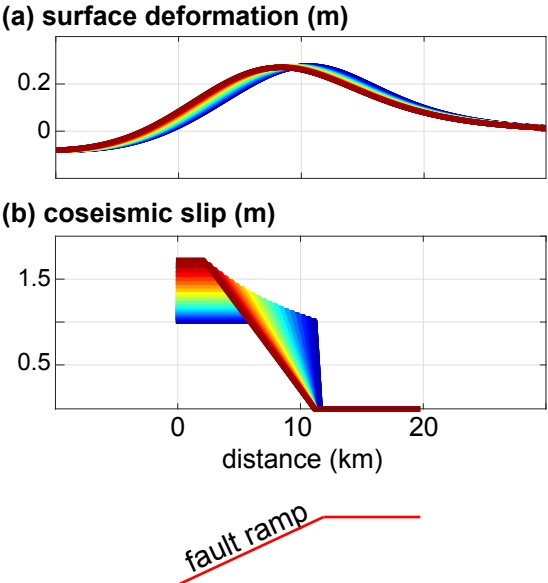


**Figure 4: Influence of tapering the updip slip on seafloor deformation. The maximum slip varies across the models in order to preserve the mean slip. (a) The seafloor deformation profiles have similar amplitudes and shapes with slightly offset peaks, even for very significant tapers. (b) Different slip tapers considered. A more gradual taper (red shades) shifts the peak uplift in the downdip direction of the fault ramp. A more abrupt slip taper (blue shades) shifts the peak uplift towards the upper fault bend.**

To better translate the models into equivalent earthquakes, we calculate the equivalent Moment Magnitude (Mw) for each modeled event, using a rigidity of 35 GPa and 30 GPa for models A and B, respectively. These are the mean rigidities calculated from the values, presented in Sallarès and Ranero (2019) and Sallarès et al. (2021), every 1 km interval from 6 to 25 km depths for Model A, and from 6 km to 15.5 km depth for model B. Since Model A has a wider fault surface, for the same amount of slip, it produces larger magnitudes compared to Model B (Table 2). In each model, we consider only the part of the fault that lies below the Lombok Strait, since this is the part of the fault that is submarine and therefore capable of generating tsunamis. We note that an earthquake rupturing this fault segment could involve slip further along strike, either to the west (below Bali) or to the east (below Lombok, although this part of the fault recently ruptured in multiple earthquakes and is relatively less likely to slip again). Indeed, reaching 5 m of slip within the Lombok Strait alone would likely require a longer rupture, and therefore a larger magnitude than the values reported in Table 2, given known scaling relationships between fault area and coseismic slip (Thingbaijam et al., 2017).

**Table 2: Equivalent Moment Magnitudes (Mw) for Models A and B for a given slip amount. Model A ruptures the full ramp while Model B ruptures only the upper half of the ramp. Both models have a fault length of 116 km. The magnitudes here are minima, as each of these events could also include slip on the along-strike part of the fault.**

309

| | Model A<br>Fault width: 45 km | Model B<br>Fault width: 22.5 km |
| --- | --- | --- |

| Slip (m) | Mw | |
|---|---|---|
| 1 | 7.5 | 7.2 |
| 3 | 7.8 | 7.5 |
| 5 | 7.9 | 7.7 |

## 2.3 Bathymetry

Accurate modeling of tsunami wave propagation requires a high-resolution bathymetric map, especially in shallow water. By using detailed bathymetry together with a fine grid size, modelled simulations of tsunami wave heights have been shown to effectively match real near-coast waveforms (Satake, 1995). However, in many parts of the world, high resolution bathymetric data are unavailable. In general, regional tsunami studies use only one bathymetric dataset (e.g., Satake, 1988), commonly either ETOPO (https://www.ngdc.noaa.gov/mgg/global/) or GEBCO (https://www.gebco.net/), because they are publicly available and have wide coverage. However, these datasets have an artificially smooth seafloor (Marks and Smith, 2006), especially at shallow depths, because of the low density of interpolated points (e.g., Fig. 5). In local tsunami studies, the detailed seafloor morphology in shallow water is critical, since seafloor roughness in these regions has nonlinear effects on wave propagation (Wang and Power, 2011). Kulikov et al. (2016) demonstrated that tsunami propagation modeled using the GEBCO dataset results in substantial errors in the estimation of wave propagation.

We generate a high-resolution bathymetric model of the region of interest by combining water depth measurements from GEBCO with sounding measurements from the official nautical charts of Indonesia (http://hdc.pushidrosal.id/). The publicly available GEBCO dataset is provided as an interpolated raster, but also includes the original data points used for interpolation. These data points (water depths) are derived from a variety of sources, both direct (echo soundings, seismic reflection, isolated soundings, electronic navigation chart soundings) and indirect (e.g. satellite altimetry, flight-derived gravity data). Using the Type-Identifier Grid file from GEBCO, which includes the source of the depth data, we identify and extract only the water depths acquired by direct measurement (Fig. 5).

The GEBCO data in this region are concentrated along the heavily-travelled ship tracks between the islands of Bali and Lombok, and are too low resolution near the coasts to accurately model tsunami propagation and wave heights (Fig. 5a). We improve the resolution of our bathymetry by digitizing sounding data from the official nautical charts of Indonesia, which are densest in the coastal regions near the cities of Denpasar (Bali) and Mataram (Lombok) and therefore critical for modeling near-shore wave heights in these regions (Fig 5b). We also trace the coastline using the National Digital Elevation Model (DEMNAS, http://tides.big.go.id/DEMNAS/), and cross check it using satellite images from Esri World Imagery (https://www.arcgis.com/).

We combine the water depth measurements from both sources and the coastlines into a single dataset, and then interpolate the data using the 'Topo to Raster' tool in ArcGIS. This tool is based on the ANUDEM program developed by Hutchinson (1989), and generates a continuous digital elevation model based on point data that takes into account the hydrological correctness of the resulting raster. While this method was developed on the basis of subaerial water flow, it has also been used to effectively generate bathymetries for tsunami studies in

other regions (Fraser et al., 2014; Darmawan et al., 2020; Wilson and Power, 2020). We note that the shallow
shelf regions of the Lombok Strait were likely incised subaerially during the late Holocene sea-level drop
(Boekschoten et al., 2000), and their morphologies therefore likely reflect subaerial water flow processes.

We set the resolution of our interpolated raster to 30 m, as this is similar to the mean distance between the data
points along the coasts of Mataram (~27) and Denpasar (~36 m). Our final bathymetry represents a reasonable
balance between achievable accuracy at shallow depths and computational efficiency. We validate the interpolated
bathymetry by comparing its values with the source data; the mean difference in the shallow regions offshore
Mataram and Denpasar is <0.4 m.

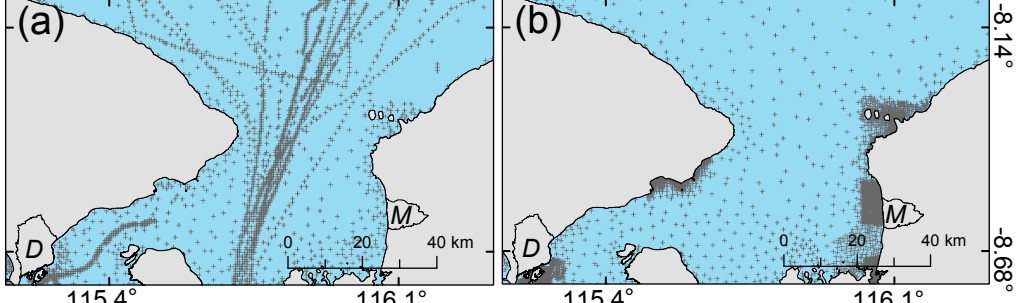


**Figure 5: Comparison of the point density of water depth measurements from (a) GEBCO (direct measurements) and**
**(b) nautical charts (soundings). GEBCO data are densest along the center of the Lombok Strait (following ship tracks),**
**while the nautical chart soundings are concentrated near the coastal cities. Combining these data points enhances the**
**accuracy of the resulting bathymetry (shown in Fig. 6). Crosses – locations of measurements. Polygons on land – cities**
**of Denpasar, Bali and Mataram, Lombok. D = Denpasar, M = Mataram.**

## 2.4 Topography in Mataram, Lombok

Based on our tsunami model runs, the highest wave heights are observed along the coast of Mataram, Lombok.
In order to further explore the tsunami hazard in this populated area (Fig. 6), we model the inundation of the
onshore region. The inundation distance and run-up height of a tsunami can vary significantly depending on
factors such as the average slope of the coast and the land cover roughness (Kaiser et al., 2011; Griffin et al.,
2015); an accurate forecast requires a high-resolution Digital Surface Model (DSM) that maps the buildings and
trees.

We use a Digital Surface Model generated by Apollo Mapping based on Pleiades satellite imagery. The DSM has
a horizontal resolution of 1.5 m and a vertical error of ±3 m. This vertical error is the lowest possible for digital
elevation models without ground control points, which we do not have access to. We use a DSM rather than a
DTM (Digital Terrain Model) to better represent the man-made structures (e.g., houses, infrastructure) present in
Mataram city. There are a few areas where the DSM is unavailable along the coast, due to difficulties in data
processing associated with tides. We fill these areas with 1.5-m resampled elevation data from DEMNAS, the
national elevation model for Indonesia, which has a coarser original horizontal resolution of 8 m. The vertical
datum of the merged data is referenced to EGM2008.

In order to run the inundation modelling, the topographic data must be merged with the bathymetry so that the
incoming wave can be smoothly modeled across the sea-land interface. To match the resolution of the DEMNAS-
DSM model, we generate another bathymetry model with 1.5 m resolution in the Mataram region using the same
'Topo to Raster' interpolation method as used previously for the bathymetry. We match the coastlines of the two
datasets to generate the final combined model.

**2.5 Tsunami modelling using COMCOT**
We model the tsunami generation, propagation, run-up and inundation using the Cornell Multi-grid Coupled
Tsunami (COMCOT) model developed by Liu et al. (1995). This modeling system solves linear and nonlinear
shallow water equations using a modified leap-frog finite difference approach (Wang & Power, 2011). It uses a
nested-grid layer algorithm to increase its computational efficiency. The Okada (1985) model is used to calculate
surface deformation due to fault slip. We use this model in our study as it has been extensively adopted and
validated for modelling tsunami events (e.g., 1960 Mw 9.5 Chilean tsunami – Liu et al. 1995; 2004 Mw 9 Indian
Ocean Tsunami - Wang and Liu, 2007; 2006 Mw 7.7 South Java tsunami – Tri Laksono et al. 2020; 2010 Mw 7.8
Mentawai earthquake – Hill et al. 2012; 2011 Tohoku tsunami – Chau and Lam, 2015).

For our tsunami modelling, we set up a total of six grid layers in a spherical coordinate system, with finer
resolution in the shallow regions along the coasts of Mataram and Denpasar (Fig. 6). For the parent grid layer
(L1), the extent covers the entire islands of Bali and Lombok (shown as the extent of Fig. 1b) and its grid size is
set to 150 m. We use 3 nested grid layers in Mataram with resolutions of 30 m (L2), 6 m (L3) and 1.5 m (L4, Fig.
6), while we use 1 sublayer in Denpasar with a grid size of 30 m (L5, Fig. 6). We added a 1.5 m grid size resolution
in Mataram to simulate the inundation of model A-5, representing the "worst case" of our various models. This
does not necessarily mean that it gives the worst-case tsunami scenario, and that a lower magnitude earthquake
can generate a comparable tsunami (Salaree et al., 2021). We only use one earthquake scenario because high
resolution inundation modeling is computationally expensive. Linear and nonlinear shallow water equations are
used on L1 and L2-L5, respectively. We set the Manning's roughness coefficient in L3-L5 to 0.013 on the water
region, and 0.03 on land (Wang and Power, 2011). The results of the simulations in grid layer L1 are shown on
Figures 7 and 8, and the results in L2 and L5 are shown on Figures 9-11. The simulations in L4 are shown as
inundation maps on Figures 12 and 13.

We run the tsunami simulation from the time of the earthquake for one hour; this is sufficient to capture both the
first wave and a series of smaller, later waves, since the coastal regions we are interested in are close to the source
(<100 km). To observe the tsunami arrival pattern along the coasts of Mataram and Denpasar within the hour, we
select virtual tide gauge locations along the 10-m bathymetric contour, facing the coastal areas where dense man-
made structures are identified from satellite images. The results of the tsunami modeling are illustrated using maps
of the initial sea surface deformation, maximum wave height, coseismic land subsidence in Bali and Lombok,
time series of wave arrivals at the virtual tide gauges, and maps of inundation depth in Mataram.


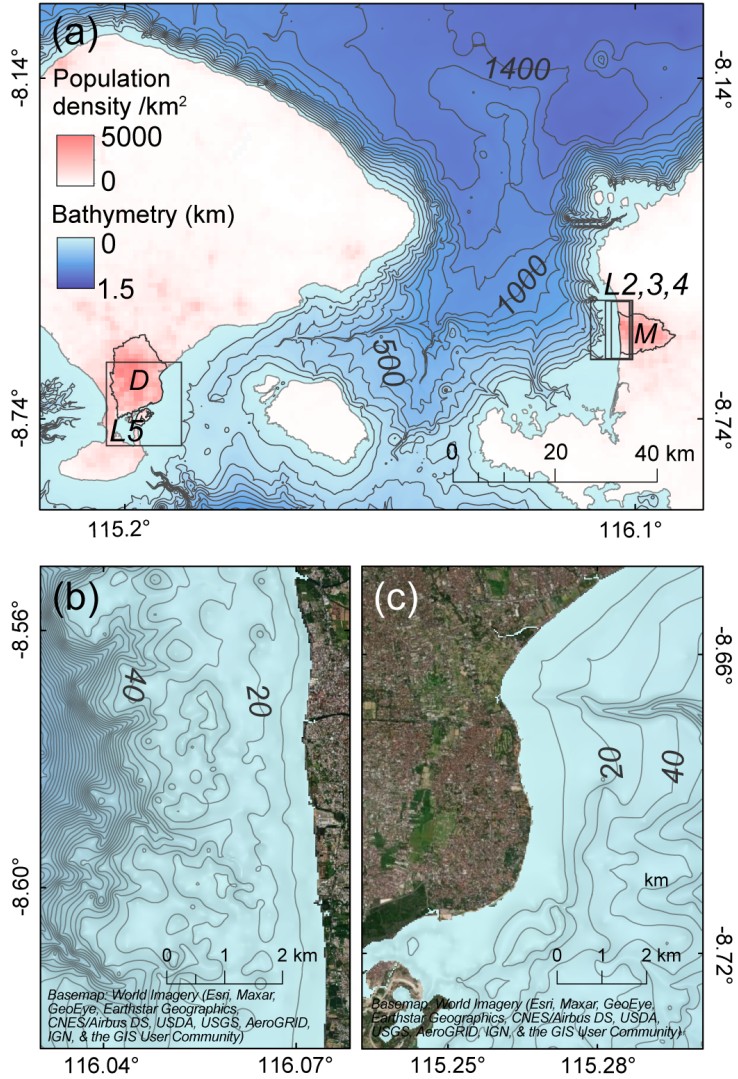

**Figure 6: The generated bathymetry in the Lombok Strait. (a) The bathymetry has a north-south trending ridge along a narrow path between Bali and Lombok with its base at 1.4 km water depth, which is the deepest water depth in this region. The extent of (a) matches the extent of grid layer L1 used in the tsunami modelling. The finer grid layers L2-L4 and L5 are focused on the populated cities of Mataram and Denpasar, respectively. M = Mataram, D = Denpasar. Grid resolutions: L1 = 150 m; L2 and L5 = 30 m; L3 = 6 m, and L4 = 1.5 m. The population density is from worldpop.org (Bondarenko et al., 2020). (b) The linear coast of Mataram faces a rugged but gently dipping seafloor that suddenly steepens ~3-4 km from the coast. (c) Denpasar city has a more complex coastline and a smoother seafloor. Basemaps – World Imagery.**

## 3 Results

### 3.1 Coseismic deformation and maximum wave height

When slip occurs on the Flores Thrust ramp during an earthquake, the elastic response of the crust will lead to broad changes in the elevation of the ground surface. In the north, above the fault ramp, the seafloor will rise (uplifting any ocean column above), whereas the southern region will subside (Fig. 7a-c). Associated with this process, the islands of Bali and Lombok will tilt towards the south (Fig. 7a-c, 8a-c). As the initial sea surface

deformation will have the same magnitude as the land deformation, the initial wave will be unnoticeable relative to the coast, which experiences the same vertical motion (Fig. 7d-f, 8d-f). As the fault patches of our fault models A (45 km) and B (22.5 km) are much larger than the ~1.4 km maximum water depth in Lombok Strait, we note that the dispersion effect due to the water column (Kajiura, 1963) is not included here. The energy transmitted to the sea surface from the seafloor by our models is only 2-3% different from the filtered versions (Felix, et. al., 2021).The initial waves in our models correspond to tsunami energies of 1, 13, and 36 TJ for Model A and 1, 7, and 20 TJ for Model B for 1, 3, and 5 m of slip, respectively (Felix et al., 2021).

The coseismic land change and tsunami heights are influenced by the distance from the fault and the shape of the coastline. Lombok and Bali have east-west trending headlands at 8.38°S latitude. In Lombok, the less protruding headland connects southwards to a north-south-trending linear coastline. In Bali, on the other hand, the headland protrudes further and connects to a southeast-facing coastline with a curved morphology. When the full fault slips (model A), the northern half of the islands, including the headlands at 8.38°S, are uplifted (Fig. 7). This uplift acts to counter any transient waves, including the initial wave, and results in a maximum relative wave height of generally <0.5 m along the northern coasts. The exception is the headlands, where the waves can be much higher; here, the waves refract towards the concave coastlines, and the wave heights can reach ~1-1.9 m high for models A-3 and A-5 (Fig. 7d-e).

Along the southern coasts, on the other hand, coseismic subsidence acts to increase the relative tsunami heights. The subsidence in southern Lombok and Bali can reach as high as ~0.3-0.4 m for model A-5, ~0.1-0.25 m for model A-3, and <0.1 m for model A-1. We find that overall, the west coast of Lombok experiences higher tsunamis than the southeast coast of Bali, because it is closer to the tsunami source and the coastline is perpendicular to the source, making it more exposed to the propagating waves. The maximum tsunami height on the west coast of Lombok is ~1.8-3.7 m for models A-3 and A-5. On the other hand, the more distant and better protected southeastern coast of Bali has a maximum wave height of ~1.3-2.2 m given the same slip amount, with slightly higher waves within the semi-enclosed bays (Figs. 7d-e).

When only the upper half of the fault ramp slips (model B), the uplift patch is narrower and the subsidence region is broader, covering about three quarters of the coasts of Lombok and Bali. Unlike in model A, the headlands at 8.38°S are now within the area of subsidence (Fig. 8). This results in an increase in the relative maximum wave height at the headlands, with ~2-4 m high tsunamis for models B-3 and B-5  (Fig. 8d-e). Similarly, the west coast of Lombok is hit by ~1.7-3.4 m high tsunamis, while southeastern Bali experiences ~0.8-2 m high tsunamis for models B-3 and B-5.

The two fault models generate similar maximum wave heights along the west coast of Lombok (Fig. 9), while the tsunamis generated by model A are slightly higher than model B along the southeastern coast of Bali (Fig. 10). In both models, however, we consistently observe higher tsunami waves in Lombok compared to Bali. This difference is best observed using the virtual tide gauge records situated near the cities of Mataram and Denpasar.

**3.2 Tsunami time series in Mataram, Lombok and Denpasar, Bali**

The tide gauge records show that the tsunami arrival times in Mataram and Denpasar are insensitive to the fault
model geometries that we consider. The first and highest wave in Mataram arrives ≤9 minutes after the earthquake
and it reaches its peak at ~11 minutes, followed by a drawdown at ~15-17 minutes. Three more waves reach the
coast at ~20, ~35 and 45 minutes (1st row, Fig. 11). The first wave in Mataram is ~2.5-2.7 m high for 5 m slip (A-
5 & B-5), ~1.6-1.7 m high for 3 m slip (A-3 & B-3), and ≤0.6 m high for 1 m slip (A-1 & B-1) (Figs. 9 & 11).
The height of the second wave is ~1.9-2.5 m, ~1.2-1.5 m, ~0.4-0.5 m, respectively, for 5 m, 3 m, and 1 m slip.
The third wave is ~0.6-0.7 m high for a 5 m slip, ~0.3-0.4 m for a 3 m slip, ~0.2-0.3 m for 1 m slip. The last wave
is ~0.1-1.3, ~0.6-1, and ≤0.2 m, respectively, for 5, 3, and 1 m slips.
In Denpasar, the waves are smaller and take longer to arrive (2nd row, Fig. 11). For fault model A, the first wave
arrives at ~12-18 minutes and reaches its peak at ~30 minutes. It is followed by a drawdown at ~38 minutes and
a second wave at ~48-53 minutes. Fault model B has a similar wave pattern with model A, however, its wave
arrival times are slightly later. The first wave in model B arrives at ~23-27 minutes, followed by a drawdown at
~40 minutes and a second wave at ~52-55 minutes (Fig. 11). As Denpasar is further from the tsunami source and
has a complex coastline, its wave records are not as uniform as those along the linear coast of Mataram. For both
fault models A and B, relatively higher tsunami waves are generated within the semi-enclosed bay in the northeast
of Denpasar, while lower waves reach southwestwards along the concave coastline (Fig. 10; Gauge 4 on Fig. 11).
Although they have a similar trend, the wave heights generated by model A are slightly higher than model B. For
model A, the maximum wave heights generated are ~1.4 m (A-5), ~0.9 (A-3) and ~0.3 m (A-1). For model B, the
maximum wave heights generated are ~0.8 m (B-5), ~0.6 m (B-3) and ≤0.2 m (B-1) (Fig. 11).


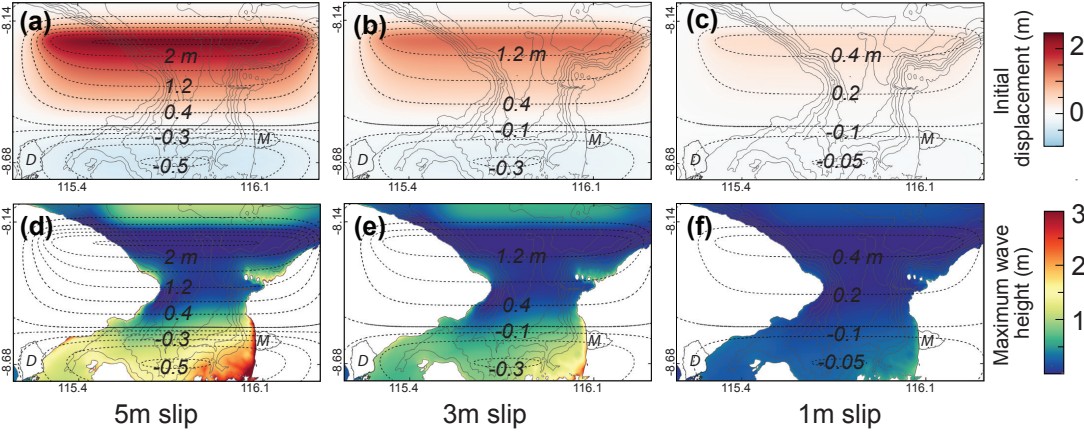


**Figure 7: Initial surface deformation and maximum wave heights in 1 hr generated by different slip amounts on the full 45-km wide fault ramp (model A). Upper panels (a-c): The coseismic deformation generated by (a) 5 m, (b) 3 m, and (c) 1 m fault slip events result in uplift in the northern half of the islands and subsidence in the south. Lower panels (d-f): Maximum sea surface displacements for (d) 5 m, (e) 3 m, and (f) 1 m fault slip events. Maps are adjusted to show wave heights relative to the post-earthquake land surface rather than initial sea level by subtracting the coseismic displacement (dashed contour lines). The west coast of Lombok is hit by higher tsunami waves than the southeastern coast of Bali. Polygons on land – cities of Denpasar, Bali and Mataram, Lombok. D = Denpasar, M = Mataram.**


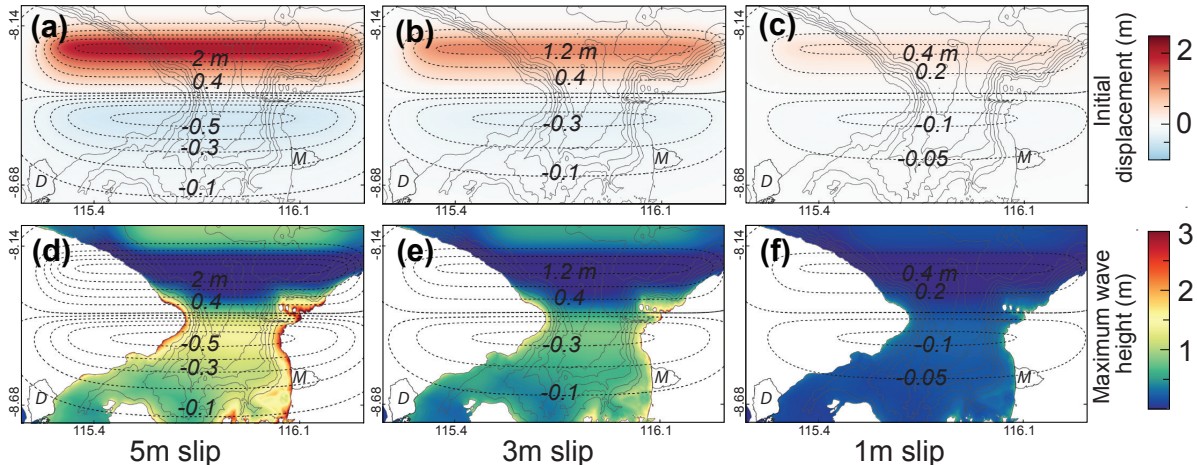

**Figure 8: Initial surface deformation and maximum wave heights in 1 hr generated by different slip amounts on the upper half of the fault ramp (model B). Upper panels (a-c): The coseismic deformation generated by (a) 5 m, (b) 3 m, and (c) 1 m fault slip events result in a narrow uplift patch in the north and broader subsidence in the south. Lower panels (d-f): Maximum sea surface displacements for (d) 5 m, (e) 3 m, and (f) 1 m fault slip events. Maps are adjusted to show wave heights relative to the post-earthquake land surface rather than initial sea level by subtracting the coseismic displacement (dashed contour lines). The highest waves are concentrated around the headlands of Lombok and Bali at 8.38°S and the mid-west coast of Lombok. Polygons on land – cities of Denpasar, Bali and Mataram, Lombok. D = Denpasar, M = Mataram.**




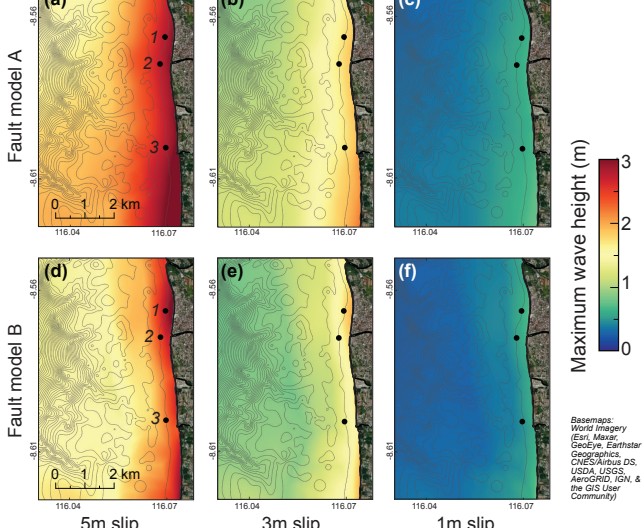

**Figure 9: Maximum wave heights in Mataram, based on simulations in grid layer 2 (L2, Fig. 6), generated by slip on fault models A (a-c) and B (d-f). Models A-5 (a) and B-5 (d) generate wave heights of ~2.5 to 2.7 m; Models A-3 (b) and**

**B-3 (e) generate ~1.6 to 1.7 m high waves; the models A-1 (c) and B-1 (f) generate ≤0.6 m high waves. Basemaps – World Imagery. Dots – tide gauges.**




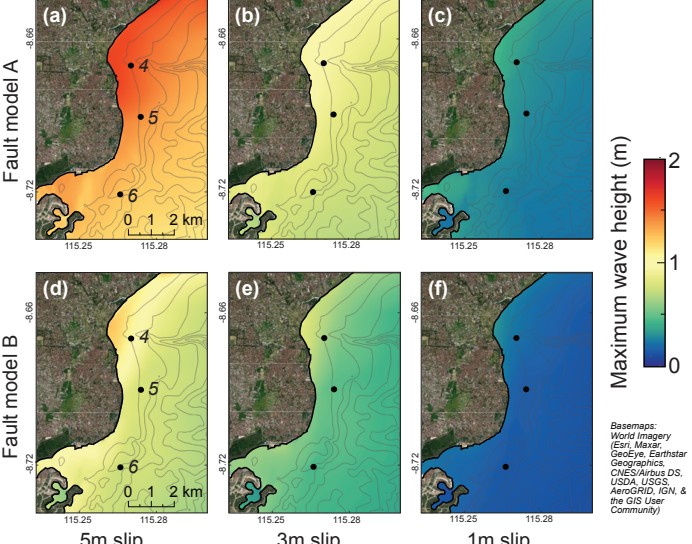

**Figure 10: Maximum wave heights in Denpasar generated by slip on fault models A (a-c) and B (d-f). The highest tsunami wave heights are located within the semi-enclosed bay on the northeast coast. The maximum wave height near Denpasar range is ~1.4 m for model A-5 (a), ~0.9 m for A-3 (b), and ~0.3 m for A-1 (c). The maximum wave heights are slightly lower in fault model B. It is ~0.8 m for model B-5 (d), ~0.6 m for B-3 (e), and ≤0.2 m for B-1 (f). Basemaps – World Imagery. Dots – tide gauges.**



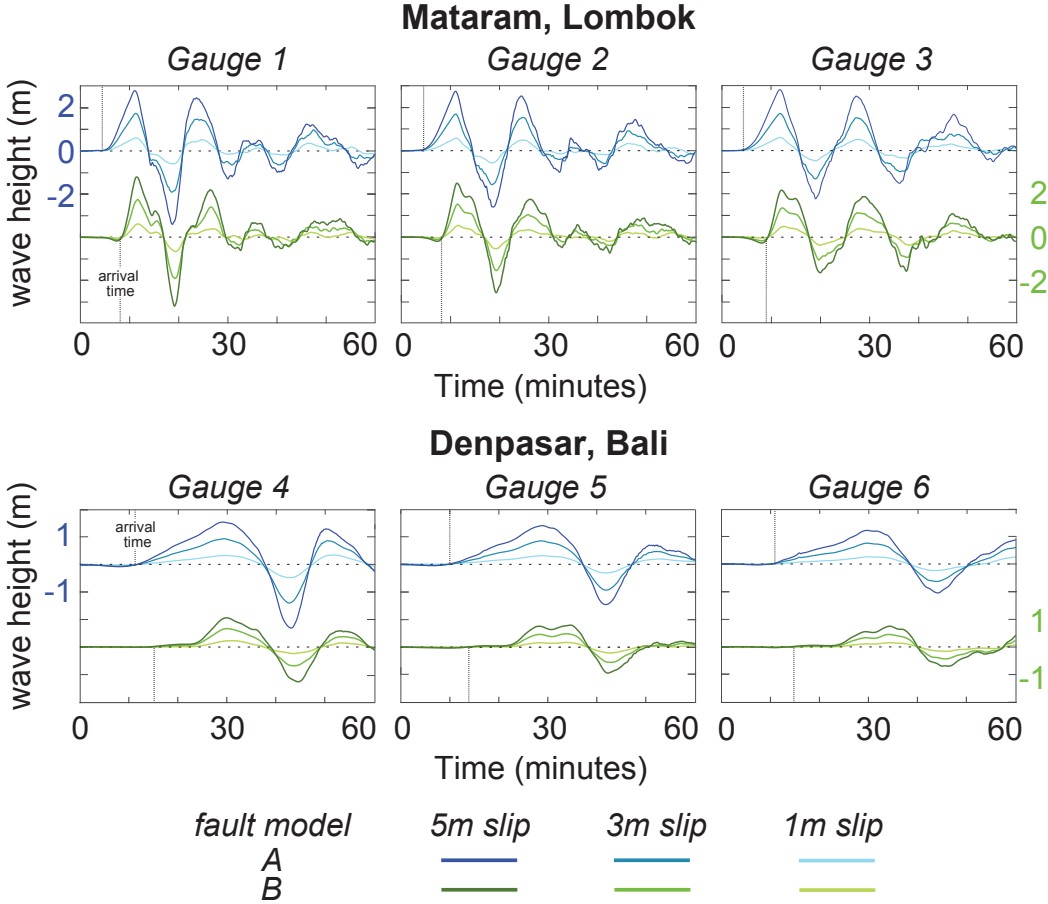

**Figure 11: Sea surface elevation generated by fault models A and B recorded at virtual tide gauges located along the 10 m water depth contours offshore Mataram (gauges 1 to 3) and Denpasar (gauges 4 to 6). The records for fault models A and B in Mataram are similar in terms of wave heights and arrival times. In Denpasar, the models have similar wave patterns but the arrival times for model A is slightly earlier than in model B. After the earthquake, the first tsunami in Mataram arrives at <9 minutes, while in Denpasar it arrives at ~12-18 minutes for model A and ~23-27 minutes. The peak of the first wave is at ~11 minutes and ~30 minutes in Mataram and Denpasar, respectively.**

### 3.3 Inundation in Mataram, Lombok

Tsunami waves of a given height at the coastline can have variable impact depending on the topography and infrastructure on land. Because inundation modeling requires a detailed Digital Surface Model for accurate results and significant computational time, we limit the inundation modeling to the city of Mataram, Lombok, because this region is densely populated (Fig. 6) and is exposed to the highest waves in our tsunami models. We run the modeling for fault model A-5 to represent the inundation of the worst-case earthquake scenario used in this study.

Based on our results, 5 m of fault slip generates two >2 m high waves followed by two lower waves that hit the coast at Mataram city (Fig. 11). These waves inundate Mataram with flow depths of generally ≤2 m but can reach as high as 3 m on the southern coast (Figs. 12 and 13). The extent of inundation is ~55-140 m along the northern to the middle parts of the coast; in the south, it reaches ~230 m. This much wider extent in the south correlates with a lower density of structures. We interpret that the presence of closely packed structures in the north limits the inundation further inland. Our results are based on the model assumption that these structures can withstand

the flow; in a real tsunami event, some structures could be destroyed (e.g., 2011 Tohoku earthquake and tsunami,
Mori et al., 2013), which could reduce flow resistance and increase the inundation distance.
The inundation has limited extent where the beach is narrow and there are dense structures near the coast. For
instance, along the northern (Figs. 12a-b) and mid-southern coasts (Figs 13a-b), inundation is limited to within
the ~15-20 m wide beach, and the closely packed residential structures just behind the beach are not inundated.
At industrial sites where there are more open spaces (Figs. 12a-b and 13a-b), the inundation extent can reach to
~95-140 m (Figs. 12a-d). When the beach is wider and the structures are further from the coast, the inundation
extends further inland (Figs. 12c-d and 13c-d). We note that in our model, clustered vegetation on the beach is
represented in the DSM as a solid barrier, and thus is able to entirely block the flow (upper part of Fig. 12c-d). In
reality, clustered vegetation can slow but not completely obstruct the flow; the inundation extent at this site is
therefore likely underestimated. Using a digital terrain model, on the other hand, would overestimate the
inundation extent (Muhari et al., 2011). Our results may be more realistic in regions where vegetation is absent,
as in the lower part of Fig. 13a-b, where we model ~175 m inundation. Along the southern coast, the beach is
generally 20-40 m wide and most of the area is farmland; with more open space, the inundation is able to reach
~230 m inland (Figs. 13c-d).

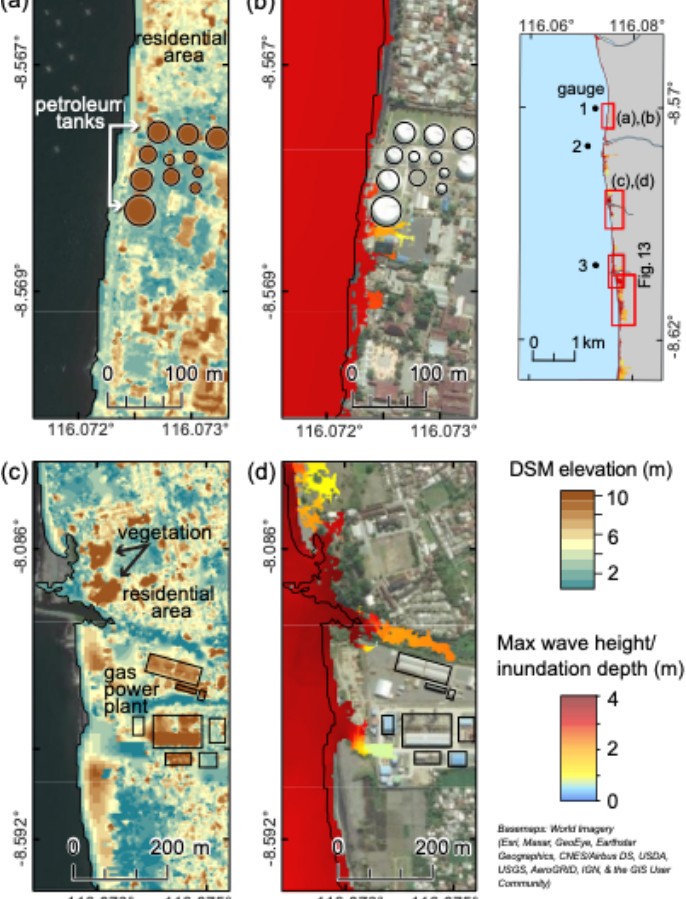


**Figure 12: The DSM elevation and inundation on the northern coast of Mataram associated with 5 m of coseismic slip on the Flores thrust ramp (model A-5) overlain on World Imagery. Flow depth is generally ≤1.5 m. (a-b) The inundation**

**extent is limited by the high density of structures in residential areas. The inundation reaches ~95 m at the industrial site (circular features are petroleum tanks), where there are more open spaces. (c-d) Inundation may be underestimated in regions where vegetation clusters act in the model as wide barriers to flow but may be more porous, as shown in the upper half of the map. In the area of the gas power plant, where there is less vegetation and the structures are more widely spaced, the inundation extent is ~140 m. Right image – location map of figures.**

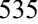


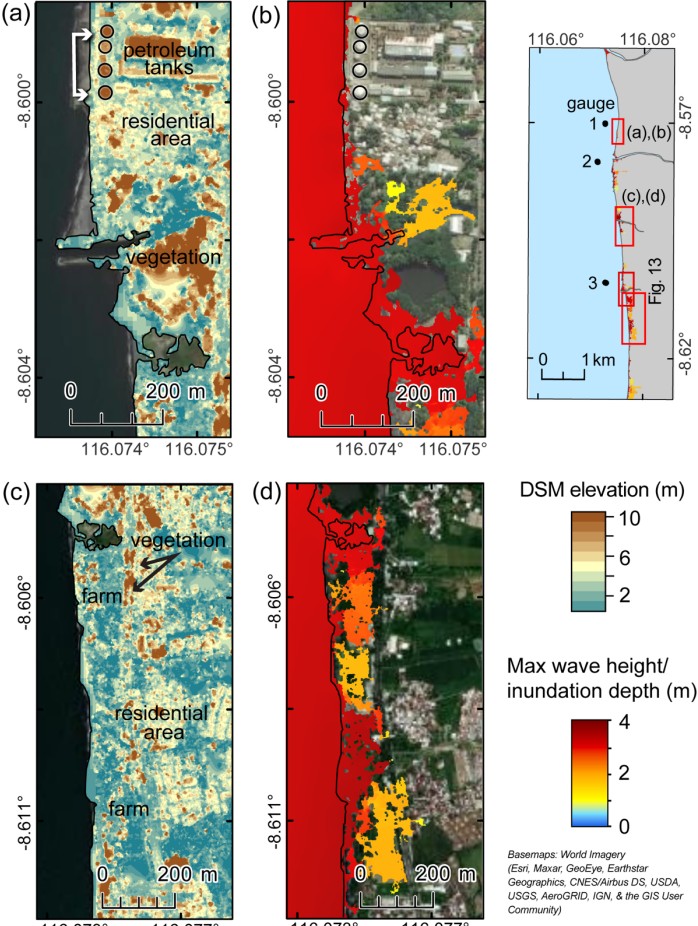


**Figure 13: The DSM elevation and inundation on the southern coast of Mataram associated with 5 m of coseismic slip on the Flores thrust ramp (model A-5) overlain on World Imagery. (a-b) To the south of the industrial site (with petroleum tanks), the inundation depth is ≤1.5 m and the inundation extent is ~175 m. (c-d) In the south, inundation is more extensive, likely because of the lower density of structures and wider open area (beach and farmland). The inundation depth is generally 2-3m and the extent reaches to ~230 m. Right image – location map of figures.**

542

## 4 Conclusions

The Flores Thrust is an active south-dipping back-arc fault system traversing north of the Lesser Sunda Islands. The 2018 Lombok earthquake sequence and prior historical events show that the western part of the fault zone is capable of generating tsunamigenic earthquakes. In this work, we study the tsunami potential associated with coseismic slip on the blind fault ramp below Lombok Strait, located between the islands of Lombok and Bali, using deterministic tsunami modelling. We focus on the tsunami patterns near the capital cities of Mataram, Lombok and Denpasar, Bali, which both lie on the coasts facing the strait. Our modeling is based on a geologically constrained model of the fault, informed by the 2018 earthquake sequence. Tsunami propagation is modeled using

a high-resolution bathymetry dataset generated by combining data points from the global GEBCO dataset with
sounding data digitized from the official nautical charts of Indonesia, interpolated using the Topo to Raster tool
in ArcGIS.

Our results show that fault rupture in this region with 1-5 m of coseismic slip could trigger a tsunami that would
hit Mataram, Lombok in ≤9 minutes and Denpasar, Bali in ~12-18 minutes with multiple waves. Furthermore,
both cities would experience coseismic subsidence of 20-40 cm, exacerbating their exposure to the tsunami hazard
and leading to more long-lasting coastal vulnerability. The maximum wave heights in Mataram are 1.6 to 2.7 m
for 3-5 m of coseismic slip, while Denpasar has maximum wave heights of 0.6 to 1.4 m. Overall, the coast along
Mataram city is more prone than Denpasar to high tsunamis arriving quickly.

Because Mataram experiences higher wave heights, we also modelled the inundation in this region for our worst-
case scenario (5 m slip) using a high-resolution DSM. We found that the inundation extends for ~55-140 m inland
with a maximum flow depth of ~2-3 m, and, except in the region just south of the city, where the inundation
reaches 230 m. This difference in inundation extent appears to be primarily influenced by the structures present
near the coast, which are denser in the north. However, if structures are destroyed by flow, inundation could reach
further inland.

Because of the proximity of the Flores thrust ramp to the coasts of Lombok and Bali, associated tsunamis would
hit within <15 minutes after the earthquake. This early tsunami arrival would mean little time for evacuation. In
the case of the 2018 Lombok earthquake, the residents of northern Lombok started evacuation only after a
government announcement, and the evacuation took at least 20 minutes (Tsimopoulou et al., 2020). For a potential
tsunami in Mataram caused by slip on the Flores thrust, there is insufficient time to wait for an announcement
after the earthquake. Hence, raising community awareness about earthquake-generated tsunamis and evacuation
plans is important, so that residents will know to respond immediately after experiencing strong ground shaking.
Furthermore, the initial polarity of the waves would be positive, and thus there would be no warning signal from
drawdown prior to inundation. In addition, a second high wave would hit Mataram coast at ~20 minutes,
emphasizing the need for continued heightened alert following the first inundation.

We finally note that some of the structures built along the coast are industrial, with several petroleum tanks and a
gas power plant. The impacts of natural disasters can be multiplied when natural events trigger industrial events
('Natural Hazards Triggering Technological Disasters,' or Natech) (Cruz and Suarez-Paba, 2019). Tsunamis in
particular have a history of causing Natech events (e.g. (Suppasri et al., 2021); for instance, the 2011 Mw9.1
Tohoku earthquake and tsunami led to not only meltdown at the Fukushima-Daichi nuclear power plant, but also
fires, explosions, and hazardous materials release at industrial sites (Krausmann and Cruz, 2013). In Mataram,
damage to the petroleum tanks, power plant, and other industrial equipment by groundshaking or inundation could
trigger Natech events, including fires, explosions, and pollution of the coastal water and associated ecological
damage. Evaluating these sites to understand and strengthen their resilience to these hazards should be a priority.

While most tsunami modeling studies in Indonesia have focused on the hazard associated with large tsunamis triggered by megathrust ruptures, such as the devastating 2004 Indian Ocean earthquake and tsunami (e.g. Wang and Liu, 2007), we highlight here the hazard associated with smaller, local events caused by slip on a back-arc thrust system. One of the challenges with local studies is the need for detailed and accurate fault models and bathymetry datasets. We show that geological information such as regional and nearby seismicity can be combined with bathymetry, topography, and seismic reflection data to model fault geometry, and that a high-resolution bathymetry dataset can be generated by combining globally available bathymetric data with sounding measurements collected for navigation purposes. Specifically, for earthquake-triggered tsunamis in Indonesia, the official nautical charts for Indonesia provide dense measurements offshore shallow coastal cities. Integrating these datasets can provide more accurate forecasts and hazard estimations for both tsunami wave height and arrival time, for local and regional studies, and could be replicated for other fault systems and areas.

**DATA AVAILABILITY**

The animation of the tsunami propagation for the 5 m coseismic slip on the full fault ramp, and the inundation model for Mataram, Lombok can be accessed freely at the Nanyang Technological University Data Repository at: https://doi.org/10.21979/N9/DZLM5D and https://doi.org/10.21979/N9/QKNSKO, respectively.

**AUTHOR CONTRIBUTION**

RPF, JAH and KEB conceptualized the research. RPF conducted the modeling and the formal analysis. JAH and KEB acquired the funding. JAH supervised the overall work. JAH, KEB and KLH assisted with the fault model setup. LL and ADS assisted with the tsunami modelling. RPF generated the figures. RPF and JAH wrote the original draft. JAH, KEB, KHL, LL and ADS reviewed and edited the manuscript.

**COMPETING INTERESTS**

The authors declare no competing interests.

**ACKNOWLEDGMENTS**

The maps in this paper were made using ArcGIS® software by Esri. The World Ocean Base map is attributed to Esri, GEBCO, NOAA, Garmin, HERE, and other contributors. The World Imagery basemap is attributed to Esri, Maxar, Earthstar Geographics, USDA FSA, USGS, Aerogrid, IGN, IGP, and the GIS User Community. The ArcGIS® and ArcMap™ are the intellectual property of Esri and are used herein under license. Copyright © Esri. All rights reserved. We would like to thank Rishav Mallick for helping in creating figure 4 using the Unicycle code (Moore et al., 2019).

This research was supported by the Earth Observatory of Singapore via its funding from the National Research Foundation Singapore and the Singapore Ministry of Education under the Research Centres of Excellence initiative. This work comprises EOS contribution number 408. The project was also supported by National Natural Science Foundation China (No 41976197).

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
