# Peer review of "Tsunami hazard in Lombok & Bali, Indonesia, due to the 2 Flores back-arc thrust"

_Natural Hazards and Earth System Sciences, 2021_

## Author Comment (AC1)

**nhess-2021-343:**
**Tsunami hazard in Lombok & Bali, Indonesia, due to the Flores backarc thrust**

The manuscript presented by Felix et al. evaluates the tsunami hazard associated with a potential rupture of the Flores backarc thrust along the Lombok Strait. The manuscript is well written, and properly organized, and figures are legible and of good quality. While modelling in this work uses higher resolution bathymetric and topographic data than previous studies, the following aspects need to be clarified/improved before considering this work for publication:

Thank you for your comments and suggestions. Please see below the list of our responses.

- The fault model is mostly interpreted on the basis of 2D seismic sections, earthquake location, and seafloor morphology (which is not clearly visible from the current figures). Yet, the region used to rupture is > 100 km wide, so one would appreciate few words regarding how potential along-strike structural variations could affect modelling results.

  The fault rupture that we consider is ~116 km wide, stretching across the strait between Bali and Lombok. While there is limited data within the strait to assess the continuity of the fault, there is no reason to believe that there are significant structural variations along strike. The focal mechanisms for the events near Bali have very similar strike and dip to that at Lombok (Fig. 1a). When varying the fault dips to 18° and 34°, representing the minimum and the maximum limits of the fault dip uncertainty, they have minimal impact on the tsunami model. The tsunami of these two models are only 5-8% different from the energy of our model with a 25° fault dip. Minor structural variations would result in minor variations in arrival times and wave heights but would not be likely to have a strong effect on our results.

  Any variations in strike and dip beyond the strait would have no impact on our results, as they would be subaerial. We will include a statement in section 2.1 (Methods – Slip model) to show that an increase in along-strike length would not affect the tsunami in the Lombok Strait, as long as the fault length is wider than the Strait's narrow opening.

- A close up of the bathymetry from Fig.1a would be needed along the Lombok Strait and North Lombok, to properly see that the inferred ramp in Fig. 1a is supported by morphological variations of the seafloor.

  We will improve the presentation of the seafloor morphology in Figure 1b (the closeup view of Lombok Strait and North Lombok) by making the contour lines thicker and darker. We will make the panels in Figure 1 bigger to increase the visibility of the minor details of the maps. A 'bare' version of Fig 1b will be included

in the supplementary material where all the overlain layers are removed, except for the contour lines.

- The potential rupture area spans along the entire Lombok Strait, yet, the 2018 sequence did not rupture through the region of the Strait, its westernmost sequence localized North Lombok (during the 5th of August) according to Lythgoe et al. (2021). So I wonder how realistic is the proposed scenarios in which the entire Lombok Strait ruptures at the same time?

  Indeed, we are not trying to replicate the 2018 earthquakes, but rather consider an earthquake on the neighboring part of the fault that did not rupture in that sequence. The eastern bound of our fault model does overlap slightly with the westernmost limit of the 2018 Lombok earthquake sequence. Such overlapping ruptures have been observed in Kuril Trench (Ammon et al., 2008) and Peru-Chile Trench (Bilek, 2010).

- The following comment concerns lines 259 to 267. The conversion to Mw from Slip assumes a standard rigidity of 30GPa (Table 1), thus assuming that rigidity is constant along the entire rupture area. In the last few years, it has become clear that rigidity of rocks overlying the fault in the upper-plate decreases trenchwards up to values of < 5 GPa in megathrust regions worldwide. Most importantly, it has been demonstrated that this variation determines shallower larger slip, longer duration and depletion of high-frequencies in the shallow thrust, conditioning tsunami wave height (Sallarès & Ranero, 2019 1038/s41586-019-1784-0; Prada et al., 2021; 10.1029/2021JB022328). Realistic rigidity variations in turn, allowed to explain the rupture of particular megathrust events (Sallarès et al., 2021 10.1126/sciadv.abg8659). Based on this, the authors should assume realistic rigidity variations with depth to provide more accurate values of Mw, assuming a constant rigidity is likely to result in incorrect estimates of Mw and slip along the rupture area (which leads me to my final comment).

  Thank you for the comment and suggested papers. We agree that rigidity, which varies with depth, is a critical issue for many earthquake studies. The low values of rigidity mentioned are indeed especially important in shallow trench regions, where low-angle thrusts cut through weak sediments. In the case of the higher angle Flores thrust, however, we do not expect these low rigidities to be a major factor, especially since the fault does not appear to reach the shallowest sediments.

  Nevertheless, we have explored the impact of rigidity variations in our case study. Using the rigidity the values presented in Sallarès & Ranero, (2019), Prada et al., (2021) and Sallarès et al., (2021), we extracted the rigidity value every one km from 6 km to 25 km depths (the depth range of the fault ramp in our study). We used these values to calculate the average rigidities of our two models. Model A, which includes the whole ramp, has an average rigidity of 35 GPa. Model B, which only includes the upper half of the ramp (from 6km to 15.5km depth), has an average of 30 GPa. Because of the change in the rigidity of Model A, we will update the calculated Mw on Table 1 and include texts after lines 259 to 267 to explain why we use these rigidity values.

| Depth (km) | Rigidity (Sallares & Ranero, 2019) | | | | |
| --- | --- | --- | --- | --- | --- |
| 6 | 22 | 21.9 | | | |
| 7 | 25 | 24.6 | Model B | 30.5 | 30.35 |
| 8 | 27 | 26.9 | Model A | 35.5 | 35.39 |
| 9 | 29 | 28.7 | | | |
| 10 | 31 | 30.2 | | | |
| 11 | 32 | 31.8 | | | |
| 12 | 33 | 33.2 | | | |
| 13 | 35 | 34.5 | | | |
| 14 | 35 | 35.4 | | | |
| 15 | 36 | 36.3 | | | |
| 16 | 38 | 37.5 | | | |
| 17 | 38 | 38.1 | | | |
| 18 | 39 | 38.8 | | | |
| 19 | 40 | 39.7 | | | |
| 20 | 40 | 40.4 | | | |
| 21 | 41 | 41.1 | | | |
| 22 | 42 | 41.5 | | | |
| 23 | 42 | 42 | | | |
| 24 | 42 | 42.5 | | | |
| 25 | 43 | 42.7 | | | |

- It is not clear to me what is the rationale behind the choice of slip values (1, 3, and 5 m). If the authors take these values based on previous estimates they should explain it. Alternatively, if the authors estimated the different slip values from Mw of previous events, they should include rigidity values used to perform the conversion. If the authors assumed a constant rigidity (as they did to convert slip into Mw), it is quite likely that they are underestimating the amount of slip that the shallow thrust may generate, and thus, the amount of uplift, and tsunami wave height. Additionally, overestimation of rigidity may also result in overestimation of tsunami arrival times, given that the rupture is likely to propagate much slower in the updip region than in the downdip.

The modelled historical tsunamigenic earthquakes in the Flores Thrust are estimated to have magnitudes ranging from Mw 6.7 to Mw 8.5 (NOAA, Musson et al., 2019; Griffin et al., 2019). Using the scaling by Thingbaijam et al. (2017), these earthquake magnitudes have average slip ranging from 1 to 5 m. In order to represent this range, we use the minimum, the mid-range and the maximum slip values in our modelling. Earthquakes with slip <1 m will result in minimal tsunamis. Earthquakes with slip >5 m are likely too large to be realistic scenarios. We will add this explanation in line 242 of section 2.1 Methodology - Slip model.

As noted previously, the rigidities in this region are unlikely to significantly impact the results, since the fault is steeply dipping and does not reach the surface (where the lowest rigidities can be found).

---

## Author Comment (AC2)

**nhess-2021-343:**
**Tsunami hazard in Lombok & Bali, Indonesia, due to the Flores backarc thrust**

The manuscript studies potential tsunami hazards in Indonesia for the east coast of Mataram (Lombok) and west coast of Denpasar (Bali). Authors apply a simple deterministic tsunami modelling to assess the tsunami hazards for those two study areas. Even tough simple, the authors utilised the most up-to-date tectonic knowledge from this region. Therefore, I see that this manuscript would provide new knowledge of tsunami hazards for this region.

The manuscript also provides short but complete summary on the regional tectonic setting, seismicity, and historical events from the past. Moreover, the authors provide a much more than adequate detail on how they build or set-up the tsunami model until they analyse the results and draw conclusion. I see that this manuscript potentially also acts as a guideline for other researchers.

In general, I recommend this manuscript to be published at NHESS after some clarification as follow:

Thank you for your comments and suggestions. Please see below the list of our responses.

Further clarification:
- [Section 2.1 Faul model setup] Authors fixed the dip angle of its fault plane model to 25°, based on the study by Lythgoe et al. (2021). Whereas it is written "26±8°" in Section 1.2 Seimicity of the Flores Thrust and there is some degree of uncertainty from the study by Lythgoe et al. (2021). Could you please qualitatively analyse what will be the impact for the tsunami model?

  Thank you for your comments. We conducted a sensitivity analysis, and determined that 18° and 34° fault dips, representing the minimum and the maximum limits of the fault dip uncertainty, have minimal impact on the tsunami model: the tsunami energies of these two models are only 5-8% different from the energy of our model with a 25° fault dip. We will add a statement about this range in the manuscript.

- [Section 2.4 Tsunami modelling using COMCOT]
  - Second paragraph, it needs many reads to understand how the authors set up the nested grid. Please clarify it. Adding the resolution on Fig 6 would help.
    We will rephrase this paragraph to improve readability. We will also add the resolution of each grid layer in the caption of Fig. 6.

  - It seems that the largest domain model (L1) does not cover all the initial sea surface deformation used. Further the L5 layer is very close to the SW corner of the boundary model leaving only a few km distance from the edge. Even though the model works fine, it is uncommon to set the targeted area (L5) like in this manuscript. The wave might be affected by the boundary domain model. Moreover, peninsula of southern west part of Lombok (about 20-30 km south of Mataram) is not fully covered by the largest domain model. It might affect on how the tsunami wave propagate from the source to the target area. I recommend to

enlarge the most-outer domain model (L1) so that can fully cover the initial sea surface deformation model and to show the result outside the targeted area. Some readers might interest to see the result, for example, for all Bali and Lombok coastline.

We will re-run our tsunami models so that the first grid layer has a wider coverage to include the entire islands of Bali and Lombok; this will ensure that its boundary is further away from the sublayers. We will update the frame of Fig 1b to reflect the boundary of grid layer 1. In the supplementary material, we will also update the tsunami animation so that the entire Bali and Lombok islands are shown. This is for the readers that are interested to know more about the tsunami impact outside the Lombok Strait.

[Section 3.3 Inundation in Mataram] A digital surface model is used from modelling the tsunami. Could you please qualitatively analyse what will be the impact if a digital terrain model is used?

A study by Muhari et al. (2011) compared the inundation in Mataram city using 5m-resolution DTM and modified DSM datasets. They showed that using a DTM (with constant manning coefficient) results in a wider inundation extent compared to a modified DSM (heights of the structures are extracted from the DSM and added to the DTM). In satellite images, Mataram city has a lot of closely packed structures; we therefore prefer the use of a digital surface model in our inundation modelling. Unlike in Muhari et al. (2011), we did not modify our DSM because it has higher resolution (1.5m) and hence, the structures are already well defined.

Suggestions:

- I recommend to assign a 'name' for each scenarios so that it would be easier to read the manuscript. For example, Fault Model A with 1 m slip = A-1; Model A with 3 m slip = A-3; etc. Then use these in the text.

  Thank you for this suggestion. We will update the text to adopt this naming convention.

- Line 395, please refer "headlands at 8.38°s" to a figure.

  We will add figure references at lines 395 (Fig. 7) and 411 (Fig. 8).

- Table 1. Please provide full parameters of these scenarios so that can be reproduced by other researchers. Parameters required such as: fault plane coordinate (centroid or a coordinate of one corner, depth), fault mechanism (strike, dip, rake).

  We will add the fault parameters in table 1.

- Figure 1. Please provide one (or two) reference(s) for the white squares.

  We will add the citations from lines 114 & 115 into the caption of Figure 1.

- Figure 9 caption, please remind reader that these are results from L4.

  We will add a sentence to explain this in the caption of figure 9.

- Figure 11. Please add a vertical line to indicate the estimated time arrival.

  We will add a vertical line at the arrival time of the first waves as suggested.

- Figure 11 caption, "Sea surface deformation generated by ...", do you mean sea water elevation?
  We will change the 'sea surface deformation' to 'sea surface elevation.'

- Figure 12. Please use the same format as other figures on how you show the coordinate.
  We will change the format of the coordinates in Figures 12 & 13 to match the other figures.

---

## Author Comment (AC3)

**nhess-2021-343:**
**Tsunami hazard in Lombok & Bali, Indonesia, due to the Flores backarc thrust**

General comments

This manuscript presents a series of tsunami scenarios in Lombok and Bali, Indonesia. The tsunamis are modeled as resulting from prospective earthquakes generated by the Flores back-arc thrust, a south-dipping thrust in the upper plate of the Java subduction zone. Tsunamis generated by other than megathrust earthquakes pose serious threats and are worth being investigated. Felix et al. make a good case of the Flores thrust as it may represent a potential threat to the Indonesian Islands as concerns should have been raised after the recent Lombok earthquake of 2018. Their work is thus of great interest and timely. However, I'm afraid that at the moment, the work suffers from a few weaknesses that need to be addressed before publication.

Thank you for your comments and suggestions. Please see below the list of our responses.

- First of all, I am afraid that the term "hazard" is a bit abused on several occasions (e.g., title, introduction line 47) so that the presented work does not meet the reader's expectations. A hazard assessment should incorporate the likelihood for future events and their consequences to happen and usually consider a large number of possible events. Instead, what has been done in this work, is the simple exploration of six tsunami scenarios.

  In the overlapping tripartite language of risk science, the key terms are hazard, vulnerability and risk. Here, we examine hazards using a scenario based approach based on the geology of the area as it is currently known. Our intention is to raise awareness of the hazard and to conduct a pilot study of the potential impacts of a tsunamigenic earthquake if one were to occur in the future. This work is not a complete tsunami hazard assessment; to do a comprehensive one would be beyond the scope of this work. In fact, a tsunami hazard study based on a probabilistic approach would require a level of geological data that is simply not available here. To make our approach and reasoning clearer, we will emphasize in the abstract, introduction and the methodology sections that we are exploring six deterministic tsunami scenarios. We note that although probabilistic approaches are becoming more common, the deterministic method is still included in recent tsunami hazard studies (e.g. Wronna et al., 2015; Roshan et al., 2016; Gonzales, et al., 2019; Escobar et al., 2020; Rashidi et al., 2020; Hussain et al. 2021; Rashidi et al., 2022).

  Our study examines hazard in the broadest sense. If the reviewer can suggest a different term that they feel encompasses the aims of the work then we would be happy to consider it. At this stage we prefer to continue to use the term hazard.

- Most importantly, however, I have concerns about the design of the earthquake rupture scenarios. Of the six rupture scenarios, only in one case (Model A/b) the dimensions (length, width, slip and magnitude) are rather consistent with one another, following fault scaling relations. Deviations from fault scaling relations are expected but should be explored systematically to estimate the variability of the resulting tsunami impacts and their usefulness for understanding the hazard. The

presented earthquake scenarios would hardly become a reference for hazard analyses or a benchmark for future in-depth studies in their present form. Also, the rupture scenarios are very simplistic (planar fault and uniform slip), and this approach's limitations are not at all discussed.

We agree that standard fault scaling relations are a useful way to ensure that fault areas and slip magnitudes are generally consistent. However, when we explored this option, we determined that this was not an effective approach for this particular fault. Since only part of the fault lies below the strait, the relationship between tsunamigenic fault area and slip becomes decoupled for larger earthquakes. We considered simply maintaining a standard relationship, but felt that it was confusing for the reader, and given uncertainties in those estimates it seemed more appropriate to us to keep the focus on slip amount rather than co-varying area and slip.

We do use a simplistic rupture scenario; we have explored the impacts of both variable fault dip and tapered slip (Fig. 4), and found that neither significantly impacted the results. The sensitivity analysis using 18° and 34° fault dips, representing the minimum and the maximum limits of the fault dip uncertainty, shows that the tsunami energies of these two models are only 5-8% different from the energy of our model with a 25° fault dip. We will add a statement about this range in the manuscript. Along-strike changes in fault geometry are possible but would likely also have limited impact, and are impossible to constrain given the available data.

Recommendations

I strongly recommend redesigning the earthquake rupture scenarios in the following way. 1) Select a range of earthquake magnitudes to explore. 2) Derive fault rupture dimensions (length, width, slip) from appropriate scaling relationships (Leonard, 2010, 2014; Thingbaijam et al., 2017). 3) Depending on your resources and objective, try different realizations around those values to explore the variability of the rupture parameters or discuss the possible implications of the variability. The occurrence of ruptures of the same size in different positions can also be explored. To explore the case of enhanced slip in the upper part of the ramp (Model B), which is worth considering, subdivide the rupture in discrete patches and apply variable slip values by conserving the mean. 4) Perform the tsunami                                                                                                                       simulations.

The use of planar fault ruptures with homogeneous slip should be then discussed in light of the potential outcomes or more realistic rupture scenarios involving a three-dimensional fault representation and heterogeneous slip distribution (see Serra et al., 2021). Is the fault model geologically well-constrained (see statement at line 205) to design more complex rupture scenarios? A justification is needed in this respect.

- Thank you for your suggestions. When there is more available information on the structural geology and the seismicity of the Flores Thrust in this region, we agree that a more complex fault model and a heterogenous slip would be a better option. However, the current information that we have about the Flores Thrust in Bali and Lombok region indicates that a simple planar structure is a good representation of the fault. Hence, we think that it is better to use a simplified fault model and uniform slip in our deterministic numerical simulation to lessen the use of random parameters that could increase the uncertainty in the results. We will add this explanation in our manuscript to support our choice of using a simplistic deterministic                                                                                        model.

   The mentioned study by Serra et al. (2021) is a great example of how high quality data can support more advanced studies. It also highlights how heterogeneous slip can impact estimated tsunami wave heights. We will add a reference to this paper

and briefly discuss how these variations, supported by new subsurface imaging, could impact estimates of tsunami waves. Although the recommendations that you have made are beyond the scope of the current study, we hope to see them implemented in the future.

- Below I add several specific comments and suggestions (sorted by line number) to improve the text up to the beginning of Chapter 3. The analysis and discussion about the tsunami simulations should be reconsidered in light of the new results.

- L4 versus L41 and all other occurrences: backarc or back-arc? I prefer the hyphenated spelling. In any case, please make a choice and stick to it.

  We will change backarc to back-arc in the title to be consistent with the main texts.

- L12-36: It would be better to rewrite the abstract without citations.

  We will rewrite L15-16 and L23-L24 to remove the citations.

- L40-45: It is unclear whether the focus is on studies on back-thrust faulting earthquakes or hazard studies incorporating seismic sources other than subduction megathrusts. Depending on the collected data, studies on specific events became available as the events occurred. It is true instead that many hazard analyses focus mainly on megathrusts, but hazard studies involving crustal earthquakes are not so rare. However, these are not the cases in the cited works. I suggest reviewing this part of the introduction by seeking inspiration from recent review papers on tsunami hazards that address this circumstance quite clearly (Behrens et al., 2021; Grezio et al., 2017).

  In L40-45, we will make it clearer that our goal is to emphasize that although it is not as common as the megathrust rupture events, the back-arc thrust ruptures are capable of generating tsunamis. We will use the suggested studies as our reference in editing the introduction.

- L88-89: How were the 29 earthquakes attributed to the Flores Thrust? Please specify if it is just because they occurred in the vicinity of the thrust or if there was a more detailed analysis.

  Aside from the vicinity, we also look at the strike and dip of the nodal planes and the depth of the seismicity to ascertain whether the earthquakes are consistent with the fault geometry of the Flores Thrust. We mention this in L90-91.

- L94: Replace "This fault system has also produced uplift on its hanging wall." with "The activity of this fault system is also testified by uplift recorded on its hanging wall."

  We will replace the text based on the suggested statement.

- L144-145: What do you mean by "observational window"? If it refers to the historical records, the observational window is short almost anywhere (Geist & Parsons, 2006; Grezio et al., 2017). That is why we rely on paleoseismological studies and other inferences on the long-term behavior of faults. Please elaborate on this statement.

  The observational window in our study refers to the historical and seismic records. To our knowledge, there are no paleo-tsunami studies in this area that are associated with the Flores Thrust. There is a paleo-deposit study in Bali, but it is

interpreted to be deposited by a tsunami generated by the megathrust rupture (Sulaeman, 2018). Hence, we rely only on historical and seismic records when we refer to a short observational window here. We will include a discussion about the limited tsunami studies related to Flores Thrust and that they are about the numerical modelling of the historical tsunamis.

- L172: Why "unrealistically large"? Please give your justification.

  We take the reviewers point on this term. We call it as unrealistically large because the most recent estimates of the magnitudes of the historical tsunamigenic earthquakes in the Flores Thrust ranges from Mw 6.6 to Mw 8.3 (Griffin et al., 2019), and that seismic records show that the 1992 Flores Island earthquake is Mw 7.9. We will add this explanation in L172. We will replace the phrase "unrealistically large" with "larger than any observed event.". This is more precise, and also doesn't make the common error of assuming that recorded history is representative of all possible behaviour.

- L179-185: I am afraid the Horspool et al. (2014) paper has been misunderstood. The model's description for the Flores thrust is the "unit sources" that are then linearly combined to form earthquake sources of all magnitudes in a magnitude-frequency distribution up to the maximum magnitudes reported in Table 1. The use of unit sources is part of a technique widely used in seismic hazard studies to save computational time. Also, what do you mean by "return period on the fault"? A return period is a quantity selected on a hazard curve for a site, it is not a property of the fault. Please reconsider all this paragraph and make sure to have understood what Horspool and coauthors did.

  We will rephrase L179-185 to show that the maximum magnitude calculated for the Flores thrust is Mw8.1, Mw8.3 and Mw8.5 for fault dips of 25-27°, and remove the Mw 6.4 earthquake equivalent for each sub fault to avoid confusion. We will rephrase the sentence about the return period to make it clear that it is about the recurrence interval of the tsunami hazard in Mataram.

- L266-267: In this statement, the cited relations do not apply. Wells and Coppersmith (1994) consider fault displacement at the surface, and only indirectly can one extrapolate the coseismic slip at depth. Hanks (2002), which is Hanks and Bakun (2002) in reality, and Hanks and Bakun (2008) focus on strike-slip earthquake ruptures, not thrusts. Biasi and Weldon's (2006) relation is about surface rupture length, not area. More recent and more appropriate scaling relations exist (Leonard, 2010, 2014; Thingbaijam et al., 2017), which would help reconsider this statement.

  We will replace the references in L266-267 with Thingbaijam et al. (2017) which has a scaling relationship for magnitude and slip of shallow crustal reverse faulting.

- L364-365: The worst-case rupture scenario does not uniquely yield the worst-case tsunami scenario at a given location (Salaree et al., 2021). Various techniques exist (Lorito et al., 2015; Volpe et al., 2019) to reduce the computational burden of inundation modeling. Please reconsider your approach or at least discuss its limitations and potential pitfalls.

  Thank you for the suggestions. As we are doing deterministic modelling, we do not use the filtering used by Lorito et al. (2015) and Volpe et al.(2019), which are designed for probabilistic tsunami modelling. We will instead discuss the limitation

of our approach by adding a statement emphasizing that the worst-case rupture scenario does not necessarily mean that it gives the worst-case tsunami scenario, and that a lower magnitude earthquake can generate a comparable tsunami (Salaree et al., 2021).

L385-387: Was any filter (Kajiura, 1963) applied to transfer the sea-bottom dislocation to the water surface? Please explain.

We did not include the filter (Kajiura, 1963) because the dispersion effect can be disregarded since the fault patches of our models (22.5km and 45km) are much larger than the ~1.4km maximum water depth in Lombok Strait. This means that the energy transmitted to the sea surface by our models is only 2-3% different from the filtered versions (Felix, et. al., 2021).

References (if not already cited in the manuscript)

Behrens, J., Løvholt, F., Jalayer, F., Lorito, S., Salgado-Gálvez, M. A., Sørensen, M., et al. (2021). Probabilistic Tsunami Hazard and Risk Analysis: A Review of Research Gaps. Frontiers in Earth Science, 9, 628772. https://doi.org/10.3389/feart.2021.628772

Geist, E. L., & Parsons, T. (2006). Probabilistic Analysis of Tsunami Hazards*. Natural Hazards, 37(3), 277–314. https://doi.org/10.1007/s11069-005-4646-z

Grezio, A., Babeyko, A., Baptista, M. A., Behrens, J., Costa, A., Davies, G., et al. (2017). Probabilistic Tsunami Hazard Analysis: Multiple Sources and Global Applications. Reviews of Geophysics, 55(4), 1158–1198. https://doi.org/10.1002/2017RG000579

Kajiura, K. (1963). The leading wave of a tsunami. Bull. Earthq. Res. Inst., 41, 535–571.

Leonard, M. (2010). Earthquake Fault Scaling: Self-Consistent Relating of Rupture Length, Width, Average Displacement, and Moment Release. Bulletin of the Seismological Society of America, 100(5A), 1971–1988. https://doi.org/10.1785/0120090189

Leonard, M. (2014). Self-Consistent Earthquake Fault-Scaling Relations: Update and Extension to Stable Continental Strike-Slip Faults. Bulletin of the Seismological Society of America, 104(6), 2953–2965. https://doi.org/10.1785/0120140087

Lorito, S., Selva, J., Basili, R., Romano, F., Tiberti, M. M., & Piatanesi, A. (2015). Probabilistic hazard for seismically induced tsunamis: accuracy and feasibility of inundation maps. Geophysical Journal International, 200(1), 574–588. https://doi.org/10.1093/gji/ggu408

Salaree, A., Huang, Y., Ramos, M. D., & Stein, S. (2021). Relative Tsunami Hazard From Segments of Cascadia Subduction Zone For M w 7.5–9.2 Earthquakes. Geophysical Research Letters, 48(16). https://doi.org/10.1029/2021GL094174

Serra, C. S., MartínezâLoriente, S., Gràcia, E., Urgeles, R., Gómez de la Peña, L., Maesano, F. E., et al. (2021). Sensitivity of Tsunami Scenarios to Complex Fault Geometry and Heterogeneous Slip Distribution: CaseâStudies for SW Iberia and NW Morocco. Journal of Geophysical Research: Solid Earth, 126(10), e2021JB022127. https://doi.org/10.1029/2021JB022127

Thingbaijam, K. K. S., Martin Mai, P., & Goda, K. (2017). New Empirical Earthquake SourceâScaling Laws. Bulletin of the Seismological Society of America, 107(5), 2225–

2246. https://doi.org/10.1785/0120170017

Volpe, M., Lorito, S., Selva, J., Tonini, R., Romano, F., & Brizuela, B. (2019). From regional to local SPTHA: efficient computation of probabilistic tsunami inundation maps addressing near-field sources. Natural Hazards and Earth System Sciences, 19(3), 455–469. https://doi.org/10.5194/nhess-19-455-2019

---

## Author Response (AR1)

**NHESS 2021-343 Response to reviewers**

**Referee #1**

The manuscript presented by Felix et al. evaluates the tsunami hazard associated with a potential rupture of the Flores backarc thrust along the Lombok Strait. The manuscript is well written, and properly organized, and figures are legible and of good quality. While modelling in this work uses higher resolution bathymetric and topographic data than previous studies, the following aspects need to be clarified/improved before considering this work for publication:

We thank the editor and all reviewers for your comments and suggestions. We have attached a pdf file that contains our responses in bullet form.

- The fault model is mostly interpreted on the basis of 2D seismic sections, earthquake location, and seafloor morphology (which is not clearly visible from the current figures). Yet, the region used to rupture is > 100 km wide, so one would appreciate few words regarding how potential along-strike structural variations could affect modelling results.

  New line 274 (in the marked-up version of the revised manuscript): We added the statement below to discuss effect of the along-strike structural variations in our modelling:

  "While there is limited data within the strait to assess the continuity of the fault, there is no reason to believe that there are significant structural variations along strike. The focal mechanisms for the events near Bali have very similar strike and dip to that at Lombok (Fig. 1a). When varying the fault dips to 18° and 34°, representing the minimum and the maximum limits of the fault dip uncertainty, they have minimal impact on the tsunami model. The tsunami energies inherent in these two models are only 5-8% different from the energy of our model with a 25° fault dip (Felix et al., 2021). Hence, minor structural variations would result in minor changes in arrival times and wave heights but would not be likely to have a strong effect on our results."

- A close up of the bathymetry from Fig.1a would be needed along the Lombok Strait and North Lombok, to properly see that the inferred ramp in Fig. 1a is supported by morphological variations of the seafloor.

  We agree with the reviewer and we have attempted to improve the presentation of the seafloor morphology in Figure 1b (the closeup view of Lombok Strait and North Lombok) by making the contour lines thicker and darker. We made the panels in Figure 1 bigger to increase the visibility of the minor details of the maps. A 'bare' version of Fig 1b is included in the supplementary material where all the overlain layers are removed, except for the contour lines. We mentioned this in the supplementary figure and in the caption of Figure 1.

- The potential rupture area spans along the entire Lombok Strait, yet, the 2018

sequence did not rupture through the region of the Strait, its westernmost sequence localized North Lombok (during the 5th of August) according to Lythgoe et al. (2021). So I wonder how realistic is the proposed scenarios in which the entire Lombok Strait ruptures at the same time?

Because of the long recurrence intervals of earthquakes on these faults, we cannot rely on recorded events to assess the full range of possible earthquakes. The events that we consider here are indeed larger and to the west of the 2018 sequence. We consider that these are realistic scenarios given that there is no data suggesting a segment boundary within the Strait. Based on your comment, we have added some additional text to the manuscript, as follows.

New line 267: We added "We are not trying to replicate the 2018 earthquakes, but rather consider an earthquake on the neighboring part of the fault that did not rupture in that sequence. The eastern boundary of the fault model slightly overlaps with the western limit of the 2018 earthquake sequence. Such overlapping ruptures have been observed in Kuril Trench (Ammon et al., 2008) and Peru-Chile Trench (Bilek, 2010)."

- The following comment concerns lines 259 to 267. The conversion to Mw from Slip assumes a standard rigidity of 30GPa (Table 1), thus assuming that rigidity is constant along the entire rupture area. In the last few years, it has become clear that rigidity of rocks overlying the fault in the upper-plate decreases trenchwards up to values of < 5 GPa in megathrust regions worldwide. Most importantly, it has been demonstrated that this variation determines shallower larger slip, longer duration and depletion of high-frequencies in the shallow thrust, conditioning tsunami wave height (Sallarès & Ranero, 2019 1038/s41586-019-1784-0; Prada et al., 2021; 10.1029/2021JB022328). Realistic rigidity variations in turn, allowed to explain the rupture of particular megathrust events (Sallarès et al., 2021 10.1126/sciadv.abg8659). Based on this, the authors should assume realistic rigidity variations with depth to provide more accurate values of Mw, assuming a constant rigidity is likely to result in incorrect estimates of Mw and slip along the rupture area (which leads me to my final comment).

Thank you for the comment and suggested papers. We agree that rigidity, which varies with depth, is a critical issue for many earthquake studies. The low values of rigidity mentioned are indeed especially important in shallow trench regions, where low-angle thrusts cut through weak sediments. In the case of the higher angle Flores thrust, however, we do not expect these low rigidities to be a major factor, especially since the fault does not appear to reach the shallowest sediments.

Nevertheless, we have explored the impact of rigidity variations in our case study. Using the rigidity values presented in Sallarès & Ranero, (2019), Prada et al., (2021) and Sallarès et al., (2021), we extracted the rigidity value every one km from 6 km to 25 km depths (the depth range of the fault ramp in our study). We used these values to calculate the average rigidities of our two models. Model A, which includes the whole ramp, has an average rigidity of 35 GPa. Model B, which only includes the upper half of the ramp (from 6km to 15.5km depth), has an

average of 30 GPa. These values are at or slightly above the value we initially used in our calculations. Likely as this crust is not part of the shallow trench system and because our fault tips out at 6 km depth, so does not cross into the regions with rigidities below ~22 GPa.

Because of the change in the rigidity of Model A, we updated the calculated Mw on Table 1 (now Table 2) and included the following text in new line 338 to explain why we use these rigidity values:

"To better translate the models into equivalent earthquakes, we calculate the equivalent Moment Magnitude (Mw) for each modeled event, using a rigidity of 35 GPa and 30 GPa for models A and B, respectively. These are the mean rigidities calculated from the values, presented in Sallarès and Ranero (2019) and Sallarès et al. (2021), every 1 km interval from 6 to 25 km depths for Model A, and from 6 km to 15.5 km depth for model B."

| Depth (km) | Rigidity | (Sallares & Ranero, 2019) | | | | |
|---|---|---|---|---|---|---|
| 6 | 22 | 21.9 | | | | |
| 7 | 25 | 24.6 | Model B | | 30.5 | 30.35 |
| 8 | 27 | 26.9 | Model A | | 35.5 | 35.39 |
| 9 | 29 | 28.7 | | | | |
| 10 | 31 | 30.2 | | | | |
| 11 | 32 | 31.8 | | | | |
| 12 | 33 | 33.2 | | | | |
| 13 | 35 | 34.5 | | | | |
| 14 | 35 | 35.4 | | | | |
| 15 | 36 | 36.3 | | | | |
| 16 | 38 | 37.5 | | | | |
| 17 | 38 | 38.1 | | | | |
| 18 | 39 | 38.8 | | | | |
| 19 | 40 | 39.7 | | | | |
| 20 | 40 | 40.4 | | | | |
| 21 | 41 | 41.1 | | | | |
| 22 | 42 | 41.5 | | | | |
| 23 | 42 | 42 | | | | |
| 24 | 42 | 42.5 | | | | |
| 25 | 43 | 42.7 | | | | |

- It is not clear to me what is the rationale behind the choice of slip values (1, 3, and 5 m). If the authors take these values based on previous estimates they should explain it. Alternatively, if the authors estimated the different slip values from Mw of previous events, they should include rigidity values used to perform the conversion. If the authors assumed a constant rigidity (as they did to convert slip into Mw), it is quite likely that they are underestimating the amount of slip that the shallow thrust may generate, and thus, the amount of uplift, and tsunami wave height. Additionally, overestimation of rigidity may also result in overestimation of tsunami arrival times, given that the rupture is likely to propagate much slower in the updip region than in the downdip.

This is an interesting point and we thank the reviewer for bring this up. There is

very little geological information about slip values of large earthquakes in this region. As such we have chosen a range of slip values that generally match the dimensions of the fault and historical earthquakes on the full range of the Flores Thrust (both here and to the east). Here we do not model slip <1 m because earthquakes with such slip, while realistic, would be unlikely to trigger significant tsunamis. We have added additional text to the manuscript to explain our reasoning.

New line 307: We added in an explanation on why we use 1m, 3m, and 5m as the slip: "The modeled historical tsunamigenic earthquakes in the Flores Thrust are estimated to have magnitudes ranging from Mw 6.7 to Mw 8.5 (NOAA, Musson et al., 2019; Griffin et al., 2019). Using the scaling relationship for magnitude and slip of shallow crustal reverse faulting by Thingbaijam et al. (2017), these earthquake magnitudes have average slip ranging from 1 to 5 m. In order to represent this range, we use the minimum (1 m), the mid-range (3 m) and the maximum (5 m) slip values in our modelling. In the subsequent texts, we refer to these slip models as A-1, A-3 and A-5 for fault model A and B-1, B-3 and B-5 for fault model B."

**Referee #2**

The manuscript studies potential tsunami hazards in Indonesia for the east coast of Mataram (Lombok) and west coast of Denpasar (Bali). Authors apply a simple deterministic tsunami modelling to assess the tsunami hazards for those two study areas. Even tough simple, the authors utilised the most up-to-date tectonic knowledge from this region. Therefore, I see that this manuscript would provide new knowledge of tsunami hazards for this region.

The manuscript also provides short but complete summary on the regional tectonic setting, seismicity, and historical events from the past. Moreover, the authors provide a much more than adequate detail on how they build or set-up the tsunami model until they analyse the results and draw conclusion. I see that this manuscript potentially also acts as a guideline for other researchers.

I general, I recommend this manuscript to be published at NHESS after some clarification as follow:

> Thank you for your comments and suggestions. Please see the attached pdf file that contains our responses in bullet form.

Further clarification:
- [Section 2.1 Faul model setup] Authors fixed the dip angle of its fault plane model to 25°, based on the study by Lythgoe et al. (2021). Whereas it is written "26±8°" in Section 1.2 Seimicity of the Flores Thrust and there is some degree of uncertainty from the study by Lythgoe et al. (2021). Could you please qualitatively analyse what will be the impact for the tsunami model?

  > Thank you for your comments.

  > New line 276 (in the marked-up version of the revised manuscript): We added the statement below:
  > "When varying the fault dips to 18° and 34°, representing the minimum and the maximum limits of the fault dip uncertainty, they have minimal impact on the tsunami model. The tsunami of these two models are only 5-8% different from the energy of our model with a 25° fault dip (Felix et al., 2021)."

- [Section 2.4 Tsunami modelling using COMCOT]
  - Second paragraph, it needs many reads to understand how the authors set up the nested grid. Please clarify it. Adding the resolution on Fig 6 would help.
    > New line 453: We replaced the old paragraph with the one shown below:

    > "For our tsunami modelling, we set up a total of six grid layers in a spherical coordinate system, with finer resolution in the shallow regions along the coasts of Mataram and Denpasar (Fig. 6). For the parent grid layer (L1), the extent covers the entire islands of Bali and Lombok (shown as the extent of Fig. 1b) and its grid size is set to 150 m. We use 3 nested grid layers in Mataram with resolutions of 30 m (L2), 6 m (L3) and 1.5 m (L4, Fig. 6), while we use 1 sublayer in Denpasar with a grid size of 30 m (L5, Fig. 6). We added a 1.5 m grid size resolution in

Mataram to simulate the inundation of model A-5, representing the "worst case" of our various models. This does not necessarily mean that it gives the worst-case tsunami scenario, and that a lower magnitude earthquake can generate a comparable tsunami (Salaree et al., 2021). We only use one earthquake scenario because high resolution inundation modeling is computationally expensive. Linear and nonlinear shallow water equations are used on L1 and L2-L5, respectively. We set the Manning's roughness coefficient in L3-L5 to 0.013 on the water region, and 0.03 on land (Wang and Power, 2011). The results of the simulations in grid layer L1 are shown on Figures 7 and 8, and the results in L2 and L5 are shown on Figures 9-11. The simulations in L4 are shown as inundation maps on Figures 12 and 13."

We also added the following text "Grid resolutions: L1 = 150 m; L2 and L5 = 30 m; L3 = 6 m, and L4 = 1.5 m" in the caption of Fig. 6.

- It seems that the largest domain model (L1) does not cover all the initial sea surface deformation used. Further the L5 layer is very close to the SW corner of the boundary model leaving only a few km distance from the edge. Even though the model works fine, it is uncommon to set the targeted area (L5) like in this manuscript. The wave might be affected by the boundary domain model. Moreover, peninsula of southern west part of Lombok (about 20-30 km south of Mataram) is not fully covered by the largest domain model. It might affect on how the tsunami wave propagate from the source to the target area. I recommend to enlarge the most-outer domain model (L1) so that can fully cover the initial sea surface deformation model and to show the result outside the targeted area. Some readers might interest to see the result, for example, for all Bali and Lombok coastline.
  Thank you for this comment. Based on your suggestions, we rebuilt the model domains and reran our tsunami models with a large coverage of the first grid layer to include the entire islands of Bali and Lombok; this ensures that the boundary of the first grid layer is further away from the sublayers. This change resulted in only a minimal change in the arrival times and maximum wave heights values. All the values are now updated in the manuscript. The results of the model runs with the larger domain are now included in the figures as suggested by the reviewer.

  We updated the frame of Fig 1b to reflect the boundary of grid layer 1 and added in the caption "The map extent of (b) reflects the coverage of grid layer 1 (L1) used in the tsunami modelling."

  In the supplementary material, we updated the tsunami animation so that all of Bali and Lombok islands are shown. This is for the readers that are interested to know more about the tsunami impact outside the Lombok Strait.

  [Section 3.3 Inundation in Mataram] A digital surface model is used from modelling the tsunami. Could you please qualitatively analyse what will be the impact if a digital terrain model is used?
  A study by Muhari et al. (2011) compared the inundation in Mataram city using 5m-resolution DTM and modified DSM datasets. They showed that using a DTM (with constant manning coefficient) results in a wider inundation extent compared to a modified DSM (heights of the structures are extracted from the DSM and

added to the DTM). In satellite images, Mataram city has a lot of closely packed structures; we therefore prefer the use of a digital surface model in our inundation modelling. Unlike in Muhari et al. (2011), we did not modify our DSM because it has higher resolution (1.5m) and hence, the structures are already well defined.

New line 661: We added "Using a digital terrain model, on the other hand, would overestimate the inundation extent (Muhari et al., 2011)."

Suggestions:
- I recommend to assign a 'name' for each scenarios so that it would be easier to read the manuscript. For example, Fault Model A with 1 m slip = A-1; Model A with 3 m slip = A-3; etc. Then use these in the text.
  Thank you for this suggestion. We adopt this naming convention.

  New line 311: We introduced the terminologies.
  "In the subsequent texts, we refer to these slip models as A-1, A-3 and A-5 for fault model A and B-1, B-3 and B-5 for fault model B."

- Line 395, please refer "headlands at 8.38°s" to a figure.
  New line 516: We referenced this text to figure 7.
  New line 531: We also referenced figure 8 when we talked about the headlands on that paragraph.

- Table 1. Please provide full parameters of these scenarios so that can be reproduced by other researchers. Parameters required such as: fault plane coordinate (centroid or a coordinate of one corner, depth), fault mechanism (strike, dip, rake).
  New line 266: We added "The complete parameters are listed in Table 1."
  New line 281: We added the table shown below as Table 1, and set the other table as table 2.

  "Table 1: Parameters of fault models A and B used in the numerical modelling."

| Parameters | Fault model A | Fault model B |
|---|---|---|
| Epicenter longtitude | 115.77° E | 115.77° E |
| Epicenter latitude | 8.3821° S | 8.2905° S |
| Focal depth | 15.5 km | 10.8 km |
| Width | 45 km | 22.5 km |
| Length | 116 km | |
| Strike | 90° E | |
| Dip | 25° S | |
| Rake | 90° | |

- Figure 1. Please provide one (or two) reference(s) for the white squares.

We added the citations from line 140 into the caption of Figure 1.
Below is the text added in the figure caption:
"(white rectangles; www.ngdc.noaa.gov; Hamzah et al., 2000; Rastogi and Jaiswal, 2006; Musson, 2012; Nguyen et al., 2015; Griffin et al., 2019). "

- Figure 9 caption, please remind reader that these are results from L4.
  We added " based on simulations in grid layer 2 (L2, Fig. 6)" in the caption of figure 9.

- Figure 11. Please add a vertical line to indicate the estimated time arrival.
  We added a vertical line at the arrival time of the first waves as suggested.

- Figure 11 caption, "Sea surface deformation generated by ...", do you mean sea water elevation?
  We changed the 'sea surface deformation' to 'sea surface elevation.'

- Figure 12. Please use the same format as other figures on how you show the coordinate.
  We changed the format of the coordinates in Figures 12 & 13 to match the other figures.

**Referee #3**

General comments

This manuscript presents a series of tsunami scenarios in Lombok and Bali, Indonesia. The tsunamis are modeled as resulting from prospective earthquakes generated by the Flores back-arc thrust, a south-dipping thrust in the upper plate of the Java subduction zone. Tsunamis generated by other than megathrust earthquakes pose serious threats and are worth being investigated. Felix et al. make a good case of the Flores thrust as it may represent a potential threat to the Indonesian Islands as concerns should have been raised after the recent Lombok earthquake of 2018. Their work is thus of great interest and timely. However, I'm afraid that at the moment, the work suffers from a few weaknesses that need to be addressed before publication.

First of all, I am afraid that the term "hazard" is a bit abused on several occasions (e.g., title, introduction line 47) so that the presented work does not meet the reader's expectations. A hazard assessment should incorporate the likelihood for future events and their consequences to happen and usually consider a large number of possible events. Instead, what has been done in this work, is the simple exploration of six tsunami scenarios.

The reviewer raises an interesting point. In the overlapping tripartite language of risk science, the key terms are hazard, vulnerability and risk. Here, we examine hazards using a scenario based approach based on the geology of the area as it is currently known. Our intention is to raise awareness of the hazard and to conduct a pilot study of the potential impacts of a tsunamigenic earthquake if one were to occur in the future. This work is not a complete tsunami hazard assessment; to do a comprehensive one would involve efforts considerably beyond the scope of this work. In fact, a tsunami hazard study based on a probabilistic approach would likely require a level of geological data that is simply not available here. To make our approach and reasoning clearer, we emphasized in the abstract (new line 21 in the marked-up version of the revised manuscript), introduction (new line 57) and the methodology sections that we are exploring six deterministic tsunami scenarios (new line 306).

The texts added are as follows:

New line 20: We assess modeled tsunami patterns generated by fault slip in six earthquake scenarios (slip of 1-5 m, representing Mw 7.2-7.9+) *"using deterministic modelling"*, with a focus on impacts on the capital cities of Mataram, Lombok and Denpasar, Bali, which lie on the coasts facing the strait.

New line 57: Here, we assess the *"deterministic tsunami hazard"* associated with the westernmost segment of the Flores Thrust

New line 313: "We note that although modelling with more complex rupture scenarios would perhaps be a more detailed option (e.g. Serra et al., 2021), the current information that we have about the Flores Thrust in Bali and Lombok region, however, is limited. Hence, we think that it is better to use a planar fault model and uniform slip to reduce the use of random parameters that could increase the uncertainty in the results. We also note

that although probabilistic approaches are becoming more common, the deterministic method is still included in many recent tsunami hazard studies (e.g. Wronna et al., 2015; Roshan et al., 2016; Gonzales, et al., 2019; Escobar et al., 2020; Rashidi et al., 2020; Hussain et al. 2021; Rashidi et al., 2022)."

Our study examines hazard in the broadest sense. If the reviewer can suggest a different term that they feel encompasses the aims of the work then we would be happy to consider it. At this stage we prefer to continue to use the term hazard.

Most importantly, however, I have concerns about the design of the earthquake rupture scenarios. Of the six rupture scenarios, only in one case (Model A/b) the dimensions (length, width, slip and magnitude) are rather consistent with one another, following fault scaling relations. Deviations from fault scaling relations are expected but should be explored systematically to estimate the variability of the resulting tsunami impacts and their usefulness for understanding the hazard. The presented earthquake scenarios would hardly become a reference for hazard analyses or a benchmark for future in-depth studies in their present form. Also, the rupture scenarios are very simplistic (planar fault and uniform slip), and this approach's limitations are not at all discussed.

We agree that standard fault scaling relations are a useful way to ensure that fault areas and slip magnitudes are generally consistent. However, when we explored this option, we determined that this was not an effective approach for this particular fault. Since only part of the fault lies below the strait, the relationship between tsunamigenic fault area and slip becomes decoupled for larger earthquakes. We considered simply maintaining a standard relationship, but felt that it was likely to be confusing for the reader, and given uncertainties in those estimates it seemed more appropriate to us to keep the focus on slip amount rather than co-varying area and slip.

We do use a simplistic rupture scenario; we have explored the impacts of both variable fault dip and tapered slip (Fig. 4), and found that neither significantly impacted the results. The sensitivity analysis using 18° and 34° fault dips, representing the minimum and the maximum limits of the fault dip uncertainty, shows that the tsunami energies of these two models are only 5-8% different from the energy of our model with a 25° fault dip. We added a statement about this range in the manuscript in new line 278. Along-strike changes in fault geometry are possible but would likely also have limited impact, and are impossible to constrain given the available data.

Recommendations

I strongly recommend redesigning the earthquake rupture scenarios in the following way. 1) Select a range of earthquake magnitudes to explore. 2) Derive fault rupture dimensions (length, width, slip) from appropriate scaling relationships (Leonard, 2010, 2014; Thingbaijam et al., 2017). 3) Depending on your resources and objective, try different realizations around those values to explore the variability of the rupture parameters or discuss the possible implications of the variability. The occurrence of ruptures of the same size in different positions can also be explored. To explore the case of enhanced slip in the upper part of the ramp (Model B), which is worth considering, subdivide the rupture in discrete patches and apply variable slip values by conserving the mean. 4) Perform the tsunami simulations.

The use of planar fault ruptures with homogeneous slip should be then discussed in light of the potential outcomes or more realistic rupture scenarios involving a three-dimensional fault representation and heterogeneous slip distribution (see Serra et al., 2021). Is the

fault model geologically well-constrained (see statement at line 205) to design more complex rupture scenarios? A justification is needed in this respect.

Thank you for your suggestions.

The mentioned study by Serra et al. (2021) is a great example of how high quality data can support more advanced studies. It also highlights how heterogeneous slip can impact estimated tsunami wave heights. We added a reference to this paper and briefly discuss how these variations, supported by new subsurface imaging, could impact estimates of tsunami waves. Although the recommendations that you have made are beyond the scope of the current study, we hope to see them implemented in the future.

New line 270: We added "We note that although modelling with more complex rupture scenarios would perhaps be a more detailed option (e.g. Serra et al., 2021), the current information that we have about the Flores Thrust in Bali and Lombok region, however, is too limited. Hence, we think that it is better to use a planar fault model and uniform slip to lessen the use of random parameters that could increase the uncertainty in the results."

Below I add several specific comments and suggestions (sorted by line number) to improve the text up to the beginning of Chapter 3. The analysis and discussion about the tsunami simulations should be reconsidered in light of the new results.

L4 versus L41 and all other occurrences: backarc or back-arc? I prefer the hyphenated spelling. In any case, please make a choice and stick to it.

We changed backarc to back-arc in the title to be consistent with the main texts.

L12-36: It would be better to rewrite the abstract without citations.

We removed the citations in the abstract.

L40-45: It is unclear whether the focus is on studies on back-thrust faulting earthquakes or hazard studies incorporating seismic sources other than subduction megathrusts. Depending on the collected data, studies on specific events became available as the events occurred. It is true instead that many hazard analyses focus mainly on megathrusts, but hazard studies involving crustal earthquakes are not so rare. However, these are not the cases in the cited works. I suggest reviewing this part of the introduction by seeking inspiration from recent review papers on tsunami hazards that address this circumstance quite clearly (Behrens et al., 2021; Grezio et al., 2017).

Thank you for this suggestion. We have revised this part of the introduction.

New line 39: We added "Tsunamis sourced from back-arc thrust faulting, although not as common as megathrust tsunamis, could also result in fatalities and severe damage and destruction to structures. Such are the cases for the Mw 7.7 1991 Limon, Costa Rica (Suárez et al., 1995), Mw 7.9 1992 Flores Island, Indonesia, and Mw 7.5 1999 Ambrym Island of Vanuatu (Regnier et al., 2003) earthquakes. Understanding the tsunami hazard associated with back-arc thrusting is therefore important. Several studies have recognized the contribution of crustal earthquakes, which includes the back-arc thrusting, in the development of tsunami hazard assessments (Selva et al., 2016; Grezio et al., 2017; Behrens et al., 2021)."

L88-89: How were the 29 earthquakes attributed to the Flores Thrust? Please specify if it is just because they occurred in the vicinity of the thrust or if there was a more detailed analysis.

Aside from the location, we also look at the strike and dip of the nodal planes and the depth of the seismicity to ascertain whether the earthquakes are consistent with the fault geometry of the Flores Thrust. We mention this in L88-92.

L94: Replace "This fault system has also produced uplift on its hanging wall." with "The activity of this fault system is also testified by uplift recorded on its hanging wall."

New line 117: We replaced the text based on the suggested statement.

L144-145: What do you mean by "observational window"? If it refers to the historical records, the observational window is short almost anywhere (Geist & Parsons, 2006; Grezio et al., 2017). That is why we rely on paleoseismological studies and other inferences on the long-term behavior of faults. Please elaborate on this statement.

Yes, incorporating paleoseismological results can be a great way to extend the observational window for active faults. To the best of our knowledge, unfortunately, there are no paleoseismological studies of the Flores Thrust that describe large earthquakes or tsunamis. We hope that by bringing attention to this hazard, more studies of the faults and its impacts may follow. We clarify this term in our revised manuscript.

New line 172: We added "Here, the observational window refers to the historical and seismic records. To our knowledge, there are no paleo-tsunami studies in this area that are associated with the Flores Thrust. There is a paleo-deposit study in Bali, but it is interpreted to be deposited by a tsunami generated by the megathrust rupture (Sulaeman, 2018). Hence, we rely only on historical and seismic records when we refer to a short observational window. The tsunami studies related to Flores Thrust are limited and they are about the numerical modelling of the historical tsunamis."

L172: Why "unrealistically large"? Please give your justification.

We agree that this phrasing was vague. We have replaced the phrase "unrealistically large" with "larger than any observed event.". This is more precise, and also doesn't make the common error of assuming that recorded history is representative of all possible behaviour

New line 206: We added "This earthquake magnitude is larger than any observed event as the most recent estimates of the historical tsunamigenic earthquakes in the Flores Thrust ranges from Mw 6.6 to Mw 8.3, (Griffin et al., 2019), and that seismic records show that the 1992 Flores Island earthquake is Mw 7.9."

L179-185: I am afraid the Horspool et al. (2014) paper has been misunderstood. The model's description for the Flores thrust is the "unit sources" that are then linearly combined to form earthquake sources of all magnitudes in a magnitude-frequency distribution up to the maximum magnitudes reported in Table 1. The use of unit sources is part of a technique widely used in seismic hazard studies to save computational time. Also, what do you mean by "return period on the fault"? A return period is a quantity selected on a hazard curve for a site, it is not a property of the fault. Please reconsider all

this paragraph and make sure to have understood what Horspool and coauthors did.

We rephrased this paragraph, removing the texts about the Mw 6.4 earthquake equivalent for each sub fault to avoid confusion.

New line 216: We added "The maximum magnitude calculated for the Flores thrust is Mw 8.1, Mw 8.3 and Mw 8.5 for fault dips of 25-27°."

New line 219: We added "They showed that for a 500-year return period, the tsunami hazard in Mataram is 10-30% most likely due to the shallow part of the Flores Thrust."

L266-267: In this statement, the cited relations do not apply. Wells and Coppersmith (1994) consider fault displacement at the surface, and only indirectly can one extrapolate the coseismic slip at depth. Hanks (2002), which is Hanks and Bakun (2002) in reality, and Hanks and Bakun (2008) focus on strike-slip earthquake ruptures, not thrusts. Biasi and Weldon's (2006) relation is about surface rupture length, not area. More recent and more appropriate scaling relations exist (Leonard, 2010, 2014; Thingbaijam et al., 2017), which would help reconsider this statement.

New line 309: We replaced the references with Thingbaijam et al. (2017) which has a scaling relationship for magnitude and slip of shallow crustal reverse faulting.

L364-365: The worst-case rupture scenario does not uniquely yield the worst-case tsunami scenario at a given location (Salaree et al., 2021). Various techniques exist (Lorito et al., 2015; Volpe et al., 2019) to reduce the computational burden of inundation modeling. Please reconsider your approach or at least discuss its limitations and potential pitfalls.

Thank you for the suggestions. As we are doing deterministic modelling, we do not use the filtering used by Lorito et al. (2015) and Volpe et al.(2019), which are designed for probabilistic tsunami modelling. We instead discuss the limitation of our approach.

New line 458: We added "This does not necessarily mean that it gives the worst-case tsunami scenario, and that a lower magnitude earthquake can generate a comparable tsunami (Salaree et al., 2021)."

L385-387: Was any filter (Kajiura, 1963) applied to transfer the sea-bottom dislocation to the water surface? Please explain.

We did not include the filter (Kajiura, 1963) because the dispersion effect can be disregarded as the fault patches of our models are much wider than the maximum water depth in our study area. However, we calculate the impact of ocean dispersion and now include this value in the text.

New line: 506: We added "As the fault patches of our fault models A (45 km) and B (22.5 km) are much larger than the ~1.4 km maximum water depth in Lombok Strait, we note that the dispersion effect (Kajiura, 1963) due to the water column is not included here. The energy transmitted to the sea surface from the seafloor by our models is only 2-3% different from the filtered versions (Felix, et. al., 2021)."

References (if not already cited in the manuscript)

Behrens, J., Løvholt, F., Jalayer, F., Lorito, S., Salgado-Gálvez, M. A., Sørensen, M., et al. (2021). Probabilistic Tsunami Hazard and Risk Analysis: A Review of Research Gaps. Frontiers in Earth Science, 9, 628772. https://doi.org/10.3389/feart.2021.628772

Geist, E. L., & Parsons, T. (2006). Probabilistic Analysis of Tsunami Hazards*. Natural Hazards, 37(3), 277–314. https://doi.org/10.1007/s11069-005-4646-z

Grezio, A., Babeyko, A., Baptista, M. A., Behrens, J., Costa, A., Davies, G., et al. (2017). Probabilistic Tsunami Hazard Analysis: Multiple Sources and Global Applications. Reviews of Geophysics, 55(4), 1158–1198. https://doi.org/10.1002/2017RG000579

Kajiura, K. (1963). The leading wave of a tsunami. Bull. Earthq. Res. Inst., 41, 535–571.

Leonard, M. (2010). Earthquake Fault Scaling: Self-Consistent Relating of Rupture Length, Width, Average Displacement, and Moment Release. Bulletin of the Seismological Society of America, 100(5A), 1971–1988. https://doi.org/10.1785/0120090189

Leonard, M. (2014). Self-Consistent Earthquake Fault-Scaling Relations: Update and Extension to Stable Continental Strike-Slip Faults. Bulletin of the Seismological Society of America, 104(6), 2953–2965. https://doi.org/10.1785/0120140087

Lorito, S., Selva, J., Basili, R., Romano, F., Tiberti, M. M., & Piatanesi, A. (2015). Probabilistic hazard for seismically induced tsunamis: accuracy and feasibility of inundation maps. Geophysical Journal International, 200(1), 574–588. https://doi.org/10.1093/gji/ggu408

Salaree, A., Huang, Y., Ramos, M. D., & Stein, S. (2021). Relative Tsunami Hazard From Segments of Cascadia Subduction Zone For M w 7.5–9.2 Earthquakes. Geophysical Research Letters, 48(16). https://doi.org/10.1029/2021GL094174

Serra, C. S., MartínezâLoriente, S., Gràcia, E., Urgeles, R., Gómez de la Peña, L., Maesano, F. E., et al. (2021). Sensitivity of Tsunami Scenarios to Complex Fault Geometry and Heterogeneous Slip Distribution: CaseâStudies for SW Iberia and NW Morocco. Journal of Geophysical Research: Solid Earth, 126(10), e2021JB022127. https://doi.org/10.1029/2021JB022127

Thingbaijam, K. K. S., Martin Mai, P., & Goda, K. (2017). New Empirical Earthquake SourceâScaling Laws. Bulletin of the Seismological Society of America, 107(5), 2225–2246. https://doi.org/10.1785/0120170017

Volpe, M., Lorito, S., Selva, J., Tonini, R., Romano, F., & Brizuela, B. (2019). From regional to local SPTHA: efficient computation of probabilistic tsunami inundation maps addressing near-field sources. Natural Hazards and Earth System Sciences, 19(3), 455–469. https://doi.org/10.5194/nhess-19-455-2019